# Near real-time $CO_2$ fluxes from CarbonTracker Europe for high resolution atmospheric modeling

Auke M. van der Woude[1,2], Remco de Kok[2,3], Naomi Smith[2], Ingrid T. Luijkx[2], Santiago Botía[4],
Ute Karstens[5], Linda Maria Johanna Kooijmans[2], Gerbrand Koren[6, 2], Harro Meijer[1],
Gert-Jan Steeneveld[2], Ida Storm[5,2], Ingrid Super[7], Bert A. Scheeren[1], Alex Vermeulen[3,5], and
Wouter Peters[2,1]

[1]Centre for Isotope Research, Energy and Sustainability Research Institute Groningen, University of Groningen, Nijenborgh
4, 9747 AG Groningen, the Netherlands
[2]Meteorology and Air Quality Group, Wageningen University, P.O. Box 47, 6700 AA Wageningen, the Netherlands
[3]ICOS ERIC, Carbon Portal, Geocentrum II, Sölvegatan 12, SE-22362 Lund, Sweden
[4]Max Planck Institute for Biogeochemistry, Hans-Knoell-Straße 10, 07745 Jena, Germany
[5]ICOS ERIC Carbon Portal, Physical Geography and Ecosystem Science, Lund University, Lund, Box 117, SE-221 00,Lund,
Sweden
[6]Copernicus Institute of Sustainable Development, Utrecht University, Utrecht, the Netherlands
[7]Department of Climate, Air and Sustainability, TNO, P.O. Box 80015, 3508 TA Utrecht, the Netherlands

**Correspondence:** Auke M. van der Woude (auke.van.der.woude@rug.nl)

**Abstract.**

We present the CarbonTracker Europe High-Resolution system that estimates carbon dioxide ($CO_2$) exchange over Europe at high-resolution (0.1 x 0.2°) and in near real-time (about 2 months latency). It includes a dynamic fossil fuel emission model, which uses easily available statistics on economic activity, energy-use, and weather to generate fossil fuel emissions with dynamic time profiles at high spatial and temporal resolution (0.1 x 0.2°, hourly). Hourly net biosphere exchange (NEE) calculated by the Simple Biosphere model Version 4 (SiB4) is driven by meteorology from the European Centre for Medium-Range Weather Forecasts (ECMWF) Reanalysis 5th Generation (ERA5) dataset. This NEE is downscaled to 0.1 x 0.2° using the high-resolution Coordination of Information on the Environment (CORINE) land-cover map, and combined with the Global Fire Assimilation System (GFAS) fire emissions to create terrestrial carbon fluxes. An ocean flux extrapolation and downscaling based on wind speed and temperature for Jena CarboScope ocean $CO_2$ fluxes is included in our product. Jointly, these flux estimates enable modeling of atmospheric $CO_2$ mole fractions over Europe.

We assess the skill of the CTE-HR $CO_2$ fluxes (a) to reproduce observed anomalies in biospheric fluxes and atmospheric $CO_2$ mole fractions during the 2018 European drought, (b) to capture the reduction of fossil fuel emissions due to COVID-19 lockdowns, (c) to match mole fraction observations at Integrated Carbon Observation System (ICOS) sites across Europe after atmospheric transport with the Transport Model, version 5 (TM5) and the Stochastic Time-Inverted Lagrangian Transport (STILT), driven by ECMWF-IFS, and (d) to capture the magnitude and variability of measured $CO_2$ fluxes in the city centre of Amsterdam (The Netherlands).

We show that CTE-HR fluxes reproduce large-scale flux anomalies reported in previous studies for both biospheric fluxes (drought of 2018) and fossil fuel emissions (COVID-19 pandemic in 2020). After applying transport of emitted $CO_2$, the

CTE-HR fluxes have lower median root mean square errors (RMSEs) relative to mole fraction observations than fluxes from a non-informed flux estimate, in which biosphere fluxes are scaled to match the global growth rate of $CO_2$ (poor-person inversion). RMSEs are close to those of the reanalysis with the data assimilation system CarbonTracker Europe (CTE). This is encouraging given that CTE-HR fluxes did not profit from the weekly assimilation of $CO_2$ observations as in CTE.

We furthermore compare $CO_2$ concentration observations at the Dutch Lutjewad coastal tower with high-resolution STILT
transport to show that the high-resolution fluxes manifest variability due to different emission sectors in summer and winter. Interestingly, in periods where synoptic scale transport variability dominates $CO_2$ concentrations variations, the CTE-HR fluxes perform similar to low-resolution fluxes (5-10x coarsened). The remaining 10% of simulated $CO_2$ mole fraction differ by $>$ 2ppm between the low-resolution and high-resolution flux representation, and are clearly associated with coherent structures ("plumes") originating from emission hotspots, such as power plants. We therefore note that the added resolution of our product
will matter most for very specific locations and times when used for atmospheric $CO_2$ modeling. Finally, in a densely-populated region like the Amsterdam city centre, our modelled fluxes underestimate the magnitude of measured eddy-covariance fluxes, but capture their substantial diurnal variations in summer- and wintertime well.

We conclude that our product is a promising tool to model the European carbon budget at a high-resolution in near real-time. The fluxes are freely available from the ICOS Carbon Portal (CC-BY-4.0) to be used for near real-time monitoring and
modeling, for example as a-priori flux product in a $CO_2$ data-assimilation system. The data is available at https://doi.org/10.18160/20Z1-AYJ2

# 1  Introduction

Anthropogenic emissions of carbon dioxide ($CO_2$) increase atmospheric $CO_2$ mole fraction levels, which contribute strongly to the increase of global temperatures (Intergovernmental Panel on Climate Change, 2021). Responses of the climate system,
including the rate of carbon exchange with the biosphere itself, are observed to change along with the unprecedented speed of $CO_2$ rise (Friedlingstein et al., 2022b). With that realisation, a total of 196 parties have pledged to the Paris Agreement which aims to reduce anthropogenic greenhouse gas (GHG) emissions and to limit global warming to 2 °C, but preferably 1.5 °C. Ratification of the Paris Agreement requires each country to set specific goals to reduce anthropogenic GHG emissions in their National Determined Commitments (NDCs), and to support independent verification of national greenhouse gas inventories
reported to the United Nations Framework Convention on Climate Change (UNFCCC).

To monitor progress towards the Paris Agreement goals and to verify reported greenhouse gas emissions, the EU is developing a monitoring and verification support (MVS) as part of their Copernicus program (Balsamo et al., 2021). The MVS is expected to heavily use observations of the carbon cycle from in-situ and satellite platforms (Pinty et al., 2017; Janssens-Maenhout et al., 2021; Balsamo et al., 2021). Quantitative use of $CO_2$ (Breón et al., 2015; Boon et al., 2016; Wang et al., 2018;
Nalini et al., 2022) and $CH_4$ (Bergamaschi et al., 2005; Henne et al., 2016; Thompson et al., 2022) mole fraction measurements or retrieval products in data-assimilation (DA) systems can provide a so-called top-down view of the European carbon balance. In such a setup, the combination of reported or simulated GHG fluxes and atmospheric transport allows a continuous compar-

ison to observations, and large discrepancies identify under- or overestimated surface fluxes. Successful implementations of this concept e.g. for Switzerland for $CH_4$ (Henne et al., 2016), halocarbons (Brunner et al., 2017) and $N_2$ and CFC (Manning et al., 2011) emissions have provided important feedback to their national emission registration entity (NER) that reports to the UNFCCC. From the MVS perspective, such efforts are best done operationally and in near real-time to quickly inform end-users (Balsamo et al., 2021).

The added value of data assimilation for atmospheric $CO_2$ mole fractions has so far mostly been on larger scales of (sub)continents, and on the biospheric or oceanic component of the carbon cycle (Peters et al., 2007; Rödenbeck et al., 2018; Monteil et al., 2020). With the current observational network mostly away from densely populated regions, this has allowed studies of regional carbon cycle anomalies such as the 2010 wildfires in Russia (Shvidenko et al., 2011; Krol et al., 2013; Guo et al., 2017), the drought of 2018 in Europe (Peters et al., 2020; Smith et al., 2020; Rödenbeck et al., 2020), and the COVID-19 crisis in 2020-2021 (Turner et al., 2020; Dou et al., 2021). For the 2018 drought, despite having large impacts on the European carbon cycle, quantification of the change in fluxes only became available about 2 years after the event (Smith et al., 2020; Ramonet et al., 2020; Thompson et al., 2022), mostly due to the burden of collecting and harmonising observational data and the time required to produce proper first-guess flux datasets to use in atmospheric data assimilation. Economic slowdowns during the recent COVID-19 crisis have spurred the development of operational fossil flux estimation systems at the country level (e.g. Liu et al. (2020); Dou et al. (2021)), pushing the community towards more timely (near real-time) and more specific (high resolution) provision of anthropogenic and biospheric carbon exchange information.

New challenges emerge for MVS at smaller scales and in fossil fuel-rich regions: the variability of biospheric fluxes is very high due to weather variability, while anthropogenic plumes from local emissions disperse quickly and mix with the regional background $CO_2$ signal. Many of the signals that MVS aims to verify therefore soon disappear in the noise of biospheric variability and weather patterns often dominate the transport of signals we observe in-situ or from space. It is difficult for MVS systems to resolve such variations and distinguish the flux signals of interest using sparse local observations, while maintaining a carbon balance that agrees with constraints offered by the integration capacity of the atmosphere over large spatiotemporal scales (Balsamo et al., 2021; Peters and Krol, 2020). Here, we make a first step towards integrating scales and constraints, as we plan to merge the existing CarbonTracker Europe large-scale system (CTE) with the high-resolution fluxes produced in near real-time (CTE-HR) we describe in this work.

Current developments in MVS target new ways to include short-term variability in the fluxes using information that is readily available in near real-time, and not necessarily based on atmospheric constraints. This can include so-called activity data which describes variations in an activity associated with carbon emissions (e.g., traffic density, industrial energy demand, or solar energy productivity). Activity data are often available with much smaller latency and at higher frequency than bottom-up emission inventories, or NER reports. Besides, emissions partly depend on meteorological conditions, on which information is available in near real-time. For example, in Super et al. (2020b) we used the relation between outside air temperature and residential heating in an optimization of 'dynamical' fossil fuel (FF) emissions for the Dutch Rijnmond area. Similar approaches are ongoing elsewhere (Guevara et al., 2021), and activity-flux relations were used to study $CO_2$ exchange during COVID-19 (Liu et al., 2020; Dou et al., 2021). For biospheric fluxes the relation between short-term $CO_2$ flux variability and

**Table 1.** Current $CO_2$ flux products. IFS: Integrated Forecast System (Agustí-Panareda et al., 2014); CT-NRT: CarbonTracker-Near Real-Time (Chen et al., 2019); EDGAR: Emission Database for Global Atmospheric Research (Janssens-Maenhout et al., 2019); CDIAC: Carbon Dioxide Information and Analysis Center (Andres et al., 1996); BFAS: Biosphere flux adjustment scheme (Agustí-Panareda et al., 2016); clim.: Climatology. ** Only column-averaged $CO_2$. Prescribed fluxes are provided for completeness but should not be used like an analysis product (Agustí-Panareda, personal communication)

|  | IFS | CT-NRT | Carbon Monitor | CTE-HR (this) |
|---|---|---|---|---|
| Lag | After calendar year | 1+ year | 2 months | 2 months |
| Resolution | 80 km | 1 ° x1 ° | National | 0.1° x 0.2° |
| Temporal resolution | Hourly | 3-hourly | Daily | Hourly |
| Fossil fuel | EDGAR Annual mean trends | CDIAC + extrapolation | Dynamic emission model | Dynamic emission model |
| Biogenic | CTESSEL + BFAS | Statistical fit | - | SiB4 + downscaling |
| Ocean | Takahashi (2009) clim. | Own clim | - | Dynamic downscaled clim. |
| Fire | GFAS | GFAS | - | GFAS |
| Data provided | Mole fractions ** | Both | national fluxes | Fluxes |
| Reference | Agustí-Panareda et al. (2014) | Jacobson et al. (2022) | Liu et al. (2020) | - |

weather are already used for example in the Vegetation Photosynthesis and Respiration model (VPRM) model (Mahadevan et al., 2008), FLUXCOM (Jung et al., 2020) and in the Carbon - Tiled ECMWF Surface Scheme for Exchange processes over Land (CTESSEL) biosphere module (Boussetta et al., 2013) of ECMWF's integrated forecast system (IFS) (Agustí-Panareda et al., 2014) and the Soil Moisture Active Passive (SMAP) mission Level 4 Carbon (L4C) (Jones et al., 2017). Including high-resolution flux variability typically enhances the skill in reproducing observed atmospheric $CO_2$ mole fractions (Mues et al., 2014; Agustí-Panareda et al., 2016), an important prerequisite for their use in an MVS.

Here, we report on the development and evaluation of a high resolution near real-time $CO_2$ flux product covering Europe. We see the use of this flux product as twofold: 1) is to rapidly gain insight in special events in the European carbon cycle such as the 2018 drought; 2) is as an easily available starting point for atmospheric modeling or data assimilation of $CO_2$ over Europe. These foreseen applications also highlight where our product differs from existing emission products (such as fossil fuel emissions from CAMS (Kuenen et al. (2022), doi:10.24380/0vzb-a387), GridFED (Jones et al., 2021) and GRACED (Dou et al., 2021)): we provide atmospheric modellers with an easy replacement of traditional bottom-up fluxes, but with recent socioeconomic- and meteorology-informed dynamic fluxes on a high spatiotemporal resolution over Europe. Our product is freely available (CC-BY-4.0) with a lag time of about 2 months behind real-time. As it also includes high-resolution fluxes from the biosphere, regridded fire emissions from the Global Fire Assimilation System (GFAS), and ocean (CarboScope-based) fluxes, it can be readily used in atmospheric modeling. Some other products that assess (parts of) the carbon cycle are shown in Table 1, which also clearly identifies the niche of our product. A third application, currently under development (but not yet achieved), will merge the CTE-HR fluxes with the existing CarbonTracker Europe fluxes, with the aim of achieving the desired additional level of consistency with large-scale constraints.

To document the fluxes and demonstrate their use, this paper is organised as follows: in Section 2 we explain how we created the different fluxes and which activity and meteorological data we use to create 0.2°x0.1° hourly fluxes. We also describe our efforts to evaluate our fluxes using atmospheric transport modeling, and analysis at district-level for Amsterdam. In Section 3 we provide (a) the flux anomalies at large scales during the 2018 European drought and COVID-19 restrictions, (b) an assessment of mole fraction residuals on regional scales through transport modeling at Integrated Carbon Observation System (ICOS) sites, and (c) a local scale comparison with fluxes measured in the city of Amsterdam. A discussion of the strengths and weaknesses, and implications of this work is given in Section 4, followed by the conclusion in Section 5.

## 2    Methods

### 2.1    High-resolution system

We create high-resolution (0.2°x0.1°, hourly) for Europe (-15°E - 35°E, 33°N - 72°N) in near real-time (2 month lag). To account for different parts of the European $CO_2$ budget, we combine multiple data-streams. The different sources that we use to estimate fossil fuel, biogenic, oceanic and wildfire $CO_2$ fluxes are summarised in Figure 1 and explained further in the following sections. The CTE-HR framework uses the CarbonTracker Data Assimilation Shell (CTDAS) framework to provide near real time flux estimates. These are not optimized with atmospheric data and can be used stand-alone, or they can later be optimized using CTE or other similar data assimilation systems.

#### 2.1.1    Anthropogenic combustion emissions

Anthopogenic emissions are highly variable in time (e.g. Nassar et al. (2013); Mues et al. (2014)), and are subject to socioeconomic factors as well as meteorology. Moreover, they are very heterogeneous in space; in this section, we elaborate on the spatio-temporal distribution of the anthropogenic combustion emissions using the GNFR sector definitions, similar to the CAMS dataset (Kuenen et al., 2022). The related data streams are highlighted in orange in Figure 1.

As a basis, we use the $CO_2$ emissions from 2017, as provided by country reports and compiled for the CAMS regional emission dataset (Kuenen et al., 2022). For all sectors except public power, these emissions are linked to activity data (e.g. amount of fuel sold). We then calculate the emission for the desired period by using activity data for that respective period (similar to (Super et al., 2020b)). Note that in this approach, we only scale activity data, and not changes in the $CO_2$ emitted per activity. The used activity data is shown in Table 2. Public power data are taken from the European Network of Transmission System Operators for Electricity (ENTSO-E). Spatial distribution is done following Kuenen et al. (2022), and summarised in Table 2 as well.

**Public power** Emissions from the public power sector (GNFR sector A) are highly variable due to political actions (e.g. moving towards/from nuclear or the use of more bio-fuel), but also social activity (e.g. Christmas eve) and meteorological variability (e.g. air-conditioning use during heatwaves) determine energy use. We take hourly generation data by production type from the ENTSO-E, (https://transparency.entsoe.eu). Generation by production type is translated to generation by fuel and

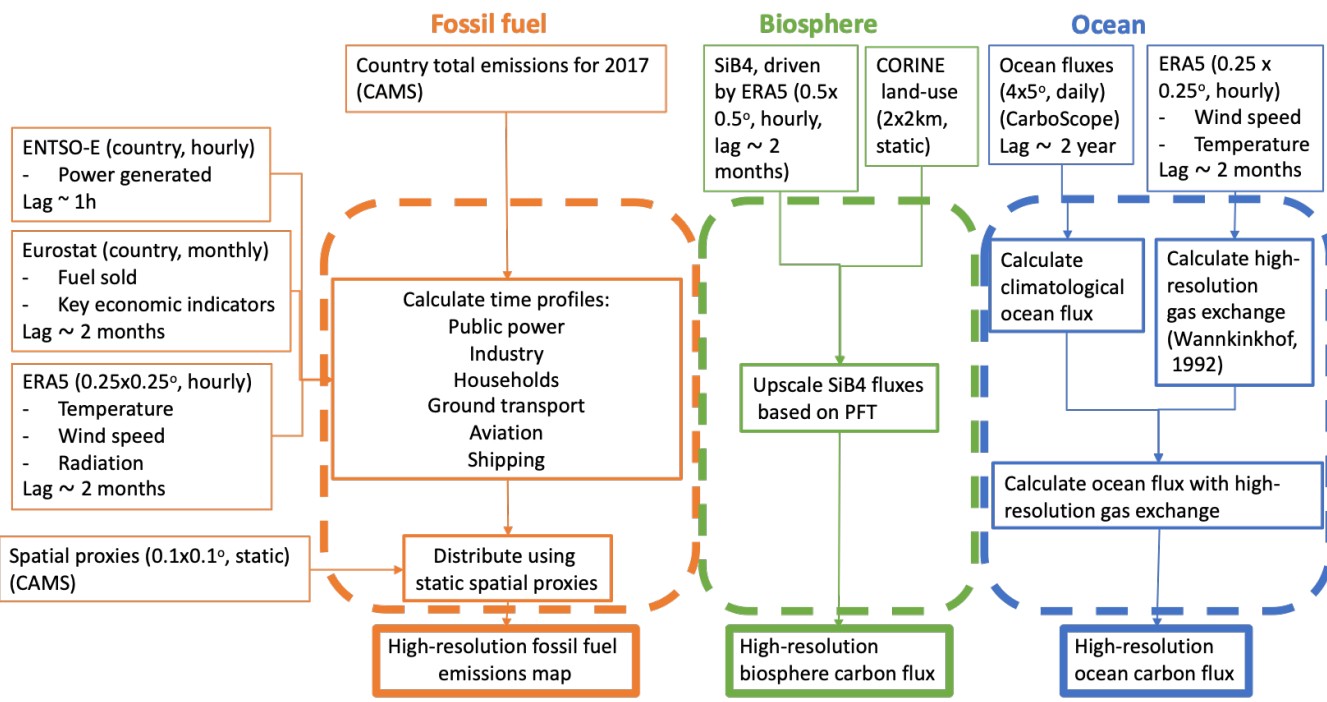

**Figure 1.** Flow chart of incoming data streams and their use in CTE-HR. The orange box shows the fossil fuel module, the blue box shows the data used for downscaling the ocean fluxes and the green box shows the calculation of the high-resolution biosphere fluxes. With all dynamic incoming data streams, the resolution and lag are shown. The individual products, such as CAMS, are described in the text.

**Table 2.** Anthopogenic combustion sectors and their spatial and temporal downscaling proxies. Dynamic indicates a dependence on meteorology, whereas the CRT (CAMS-Regional Time profiles) denotes static activity data, taken from Guevara et al. (2021). None means a flat profile (i.e. no scaling based on activity) for that time period: if a sector has a flat monthly profile, every month is assumed to have the same emissions. EU*: Eurostat data. Note that we only include surface emissions.

| (GNFR) Sector | Spatial downscaling | Activity data | | |
|---|---|---|---|---|
| | | Hourly | Daily | Monthly |
| (A) Public power | Power plant databases | Country-specific ENTSO-E generation data | | |
| (B) Industry | Point source database | van der Gon et al. (2011) | | Industry indicators (EU*) |
| (C) Other stationary combustion | Population density | CRT | Dynamic | Dynamic |
| (F) On-road emissions | Road network; Population density | CRT | CRT | Fuel demand (EU*) |
| (G) Shipping | Shipping tracks | None | None | Bunker fuel demand (EU*) |
| (H) Aviation | Airport locations | None | None | Kerosine demand (EU*) |

then translated to $CO_2$ emissions, using $E = P * C / \eta$, where $E$ denotes the $CO_2$ emissions in kg; $P$ is the energy produced in

kWh; $C$ is the carbon content of the fuel; and $\eta$ is a fuel-specific efficiency. Values for $C$ and $\eta$ for different fuels are shown in Table 3. Note that we take biofuels into account here.

With the near real-time ENTSO-E data, we capture variability in the $CO_2$ emissions of the public power sector due to variability in the amount of energy generated by renewable energy. This is shown in Section 2.2.2.

**Table 3.** Constants for electricity production from different fuels. Carbon contents are taken from Watter (2015); efficiencies from Hussy et al. (2014) and European Environment Agency (2016)

| Combustible | Carbon content $[kgCO_2/kWh]$ | Efficiency [-] |
|---|---|---|
| Biomass | 0.39 | 0.22 |
| Fossil Peat | 0.38 | 0.22 |
| Fossil Brown coal/Lignite | 0.36 | 0.38 |
| Fossil Hard coal | 0.34 | 0.38 |
| Fossil Oil | 0.26 | 0.3 |
| Fossil Oil shale | 0.26 | 0.3 |
| Fossil Gas | 0.24 | 0.47 |
| Other | 0.30 | 0.3 |

Not all countries in the European domain are included in the ENTSO-E database. Also, it can be that data is missing from the ENTSO-E database. This missing data is filled with Eurostat monthly data (https://ec.europa.eu/eurostat/databrowser/view/ NRG_CB_PEM__custom_200961/ and CAMS emissions, downscaled to hourly data if ENTSO-E data is not available for the respective country. This downscaling is achieved using the heating degree day (HDD) method and proxies for renewable energy, similar to Super et al. (2020b). For this, we assume a temperature threshold of 25 °C for coal-fired power plants (Super et al., 2020b). We also assume that the gas-fired power plants only show variability due to the variable generation of renewable energy and that oil- and biomass-fired power plants have static emissions, which is also seen in the ENTSO-E data. Power plants have a constant offset, representing generation for sources that use energy constantly. This constant offset depends on the country and month and is calculated from the available ENTSO-E and Eurostat data, respectively. If no data is available for a country, it is assumed that the offset for that country is the same as for the whole of Europe. The downscaled Eurostat data have a Pearson R of 0.5 to 0.9 compared to the ENTSO-E data, depending on the country and period tested. The country-total $CO_2$ emissions, as calculated by the HDD method and Eurostat data, differ by a maximum of 10% per week compared to the ENTSO-E data. The spatial distribution applied is the same as in Kuenen et al. (2022). Although different combustibles are included in CTE-HR, we do not differentiate between different generation units within the country; i.e. country-wide generation is projected on the relative emissions from CAMS.

**Industry** Emissions from the industry sector (GNFR sector B) are sensitive to societal and economic changes. We calculate a monthly specific scaling factor relative to 2017 for industry production volume, provided by Eurostat. The production volume is, amongst others, based on turnover of capitalised production and changes in stocks (https://ec.europa.eu/eurostat/cache/ metadata/en/sts_esms.htm). We use season and calendar adjusted data.

If the industry production data is not (yet) available for the current month, it is assumed that for that country and month, the relative production volume is equal to the average of Europe. If European data are not present, we use the previous available month for the respective country. Hourly emissions are calculated from these monthly emissions following van der Gon et al. (2011) and spatial distribution is from Kuenen et al. (2022).

**Other stationary combustion** GNFR sector C, other stationary combustion, includes household emissions, but also the commercial and institutional sectors, as well as other stationary sectors. Here, we assume that the stationary combustion $CO_2$ emissions depend mostly on outdoor temperature, as most $CO_2$ is emitted for heating. Therefore, we use the Heating Degree Day (HDD) method (Mues et al., 2014; Super et al., 2020b). We use a temperature threshold for heating of 18 °C and a constant offset (representing non-temperature dependent emissions, such as cooking) of 0.1 for all countries, similar to Mues et al. (2014); Super et al. (2020b). For a more elaborate description of the HDD method, see Mues et al. (2014).The daily emissions calculated using the HDD method are downscaled to hourly emissions using static profiles by Guevara et al. (2021). The spatial distribution of the emissions is the same as in Kuenen et al. (2022).

**On-road emissions** The on-road sector, GNFR F, includes all on-road transport, i.e. passenger cars and light and heavy duty vehicles. We do not distinguish between different sub-sectors, and therefore only account for the total $CO_2$ emissions on the road. The amount of traffic on the road is highly variable in time, therefore traffic emissions are also highly variable. Monthly petrol demand from EuroStat, relative to 2017, is used to scale the emissions (https://ec.europa.eu/eurostat/databrowser/view/nrg_jodi/default/table?lang=en). Additionally, gridded daily and hourly time factors are taken from Guevara et al. (2021). Note that these diurnal profiles are different for weekdays and weekend days, but do not include socioeconomic changes such as the COVID-19 crisis. Therefore, the diurnal cycles during e.g. the COVID-19 pandemic might differ, but this does not affect total (monthly) emissions. We apply the same spatial distribution as Kuenen et al. (2022).

**Shipping** Shipping emissions, GNFR sector G, depend highly on economic activity. We scale monthly shipping emissions with the demand of fuel oil from Eurostat, relative to 2017 (https://ec.europa.eu/eurostat/databrowser/view/nrg_jodi/default/table?lang=en). On a yearly basis, this correlates well with available CAMS emissions (not shown). The spatial distribution is assumed static and is taken from Kuenen et al. (2022).

**Aviation** Similar to shipping emissions, we scale monthly aviation emissions (GNFR sector H) based on the demand of fuel (kerosine), as supplied by Eurostat (https://ec.europa.eu/eurostat/databrowser/view/nrg_jodi/default/table?lang=en). Note that, as we use the GNFR sector definitions, only takeoff and landings are included (United Nations economic comission for Europe, 2014). We take the location of airports from Kuenen et al. (2022), and assume these locations to be static. Therefore, this does not account for newly build airports.

**Off-road emissions** Off-road emissions include tractors, construction machinery, trains and other mobile emission sources that are off-road (GNFR sector I). Currently, the CAMS emissions from the last available year are used, assuming no temporal downscaling. The spatial distribution is taken from Kuenen et al. (2022).

A summary of the relative contributions of all sectors included is shown in Table 4. The resulting $CO_2$ emissions from fossil fuels for July 8, 2018, 12h as calculated by the model are shown in Figure 2. Note that fluxes at this resolution can be created with a lag of about 2 months (see Figure 1).

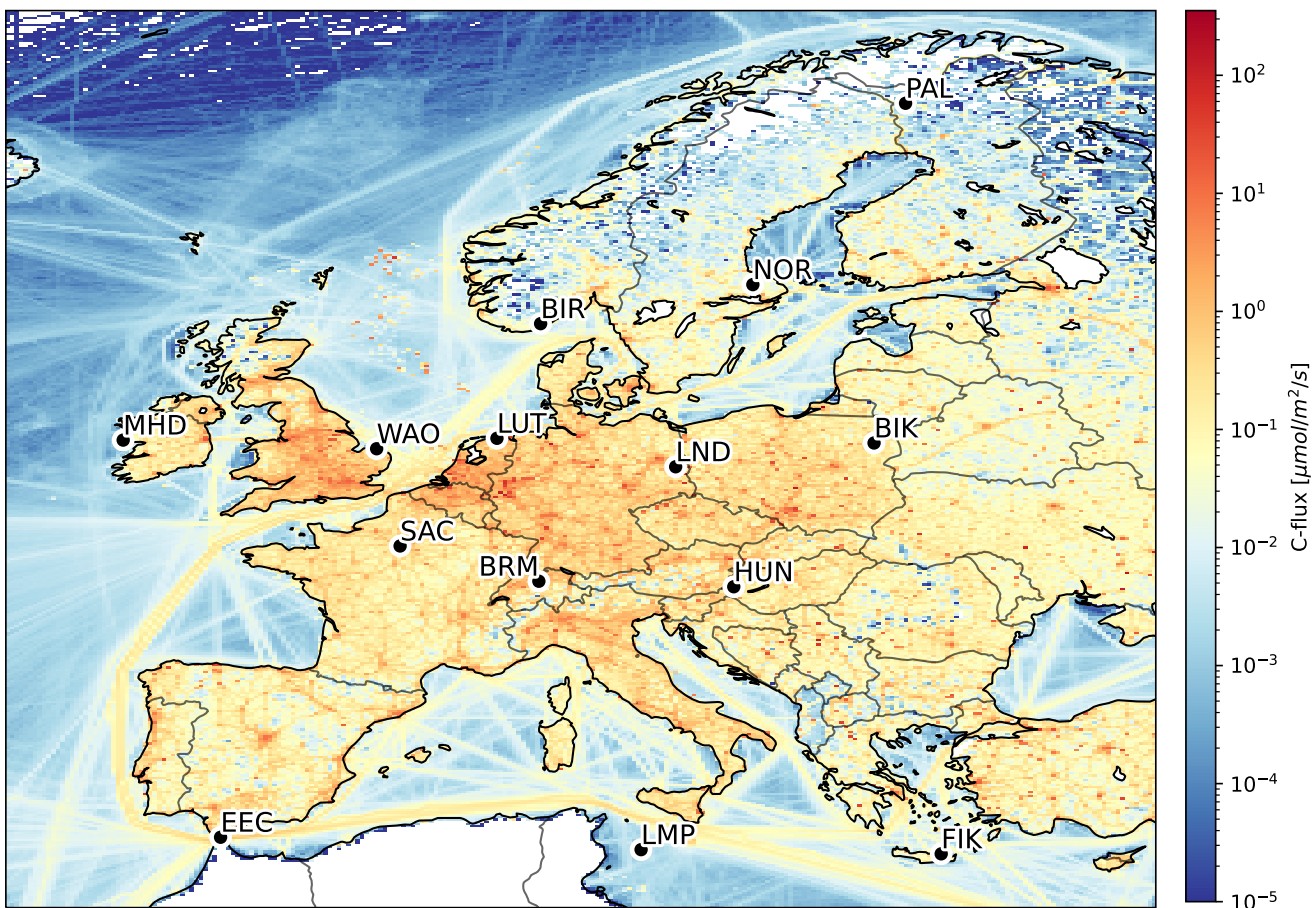

**Figure 2.** Estimated anthropogenic combustion emissions for July 8, 2018, 12h UTC as an example of the anthropogenic emisison part of our product. Black dots indicate the location of used ICOS $CO_2$ measurement sites.

### 2.1.2 Emissions from cement production

For 2018, the calcination of cement accounts for about 2% of the anthropogenic $CO_2$ emissions in the domain. We include these fluxes by taking GridFED calcination fluxes from the last available year, currently without near real-time scaling. Note that for 200 the analysis, we use Version 2021.1, but the released fluxes will contain the newest version (Version 2021.3). The carbonation of cement is currently not included. This sink accounts for about 2% of the total anthropogenic emissions (Friedlingstein et al., 2022b).

**Table 4.** Ratio of total $CO_2$ emissions from CAMS for 2017. The included column shows whether a sector is included in the final data product and included in the emissions presented in the remainder of this paper. Downscaled sectors are subject to near real-time information as described in the text.

| GNFR | Long name | Ratio of total emissions | Included | Downscaled |
|------|-----------|--------------------------|----------|------------|
| A | Public Power | 0.33 | Yes | Yes |
| B | Industry | 0.24 | Yes | Yes |
| C | Other Stationary Combustion | 0.15 | Yes | Yes |
| D | Fugitives | 0.006 | No | No |
| E | Solvents | 0.003 | No | No |
| F | On-road emissions | 0.21 | Yes | Yes |
| G | Shipping | 0.03 | Yes | Yes |
| H | Aviation | 0.004 | Yes | Yes |
| I | Off-Road | 0.03 | Yes | No |
| J | Waste | 0.0007 | No | No |
| K | Agriculture Livestock | 0.0 | No | No |
| L | Agriculture Other | 0.002 | No | No |
| | Sum of included sectors | 0.996 | | |
| | Sum of downscaled sectors | 0.934 | | |
| | Sum of all sectors | 1.0 | | |

### 2.1.3 Biosphere fluxes

Hourly biosphere fluxes inside the high-resolution domain are calculated with the Simple Biosphere Version 4 (SiB4) (Haynes
et al., 2019). SiB4 is driven by ECMWF Reanalysis 5th Generation (ERA5) meteorological input data (Hersbach et al., 2020) and restarted from a 5 x 20 years spin-up. We use a constant atmospheric $CO_2$ mole fraction of 370 ppm, resulting in a neutral steady-state biosphere. Fires are not accounted for in the spinup. Before each simulation, a 3-year run with constant $CO_2$ is used as additional spin-up, to equilibrate croplands.

To better account for water stress, we increased the drought sensitivity of evergreen needle-leaf forests and croplands by
210 modifying the rooting depth of these two plant functional types (PFTs), similar to Smith et al. (2020). Note that we do not scale precipitation from the ERA5 input using the global precipitation reanalysis product (GPCP), unlike Baker et al. (2010) did for SiB4 driven by the Modern-Era Retrospective analysis for Research and Applications Version 2 (MERRA2) meteorology, as the scaling resulted in non-physical jumps in biosphere fluxes between GPCP (2.5x2.5°) grid cells. We used unscaled precipitation from ERA5, assuming it is a reliable precipitation product for Europe.

In SiB4, the net ecosystem production (NEP) is calculated from the gross primary production (GPP) and total ecosystem respiration (TER). GPP, TER and NEP are calculated for the 10 most dominant plant functional types (PFTs) in a grid cell. We map the calculated PFT-specific fluxes at 0.5 by 0.5 degree to high resolution (up to 2km), using the coordination of information

**Table 5.** SiB4 PFT names and their corresponding CORINE land-use classes grid code. For the CORINE classification, see http://clc.gios.gov.pl/doc/clc/CLC_Legend_EN.pdf

| SiB4 name | SiB4 PFT | CORINE land-use class |
|---|---|---|
| Desert or bare ground | 1 | 1, 2, 3, 4, 5, 6, 7, 8, 9, 31 |
| Evergreen Needleleaf Forest | 2 | 24 |
| Deciduous Needleleaf Forest | 4 | - |
| Evergreen Broadleaf Forest | 5 | 15, 16, 17 |
| Deciduous Broadleaf Forest | 8 | 23, 22, 25 |
| Shrublands (Non-tundra) | 11 | 28, 29 |
| Shrublands (Tundra) | 12 | - |
| C3 plants | 13 | - |
| C3 Grass | 14 | 10, 11, 18, 26, 27, 35, 36 |
| C4 Grass | 15 | - |
| C3 Crops | 17 | 12, 14, 19, 20, 21 |
| C4 Crops | 18 | 13 |
| Maize | 20 | - |
| Soybean | 22 | - |
| Winter Wheat | 24 | - |

on the environment (CORINE) land-use classification (Bossard et al., 2000). We translated the CORINE land-use classes to SiB4 PFTs as shown in Table 5.

The effect of the high-resolution land-use map is shown in Figure 3. In the Eastern part of the domain, no CORINE data is available, and no downscaling is applied, resulting in relatively coarse spatial flux patterns . This is highlighted in the insets in panels c and d in Figure 3. See Appendix 2.2.1 for a further validation on this aspect.

### 2.1.4    Emissions from fires

Wildfire $CO_2$ emissions are taken from the Global Fire Assimilation System (GFAS) (Di Giuseppe et al., 2018). The 0.1 by 0.1
225    ° daily fluxes are binned to the 0.1 by 0.2° domain for ease of use.

### 2.1.5    Ocean fluxes

Ocean $CO_2$ exchange responds to the difference in partial pressure ($\Delta$P) of $CO_2$ in the water and atmosphere, and temperature and wind speed (Wanninkhof, 1992). Assuming constant $\Delta$P, we scale CarboScope climatological ocean fluxes of the 10 most recently available years (Rödenbeck et al., 2013) based on the gas-exchange coefficient $k$ (Wanninkhof, 1992). The
high-resolution flux is calculated as $F = F_{clim} * k/k_{clim}$, where $F$ is the ocean-atmosphere $CO_2$ flux and the subscript $clim$

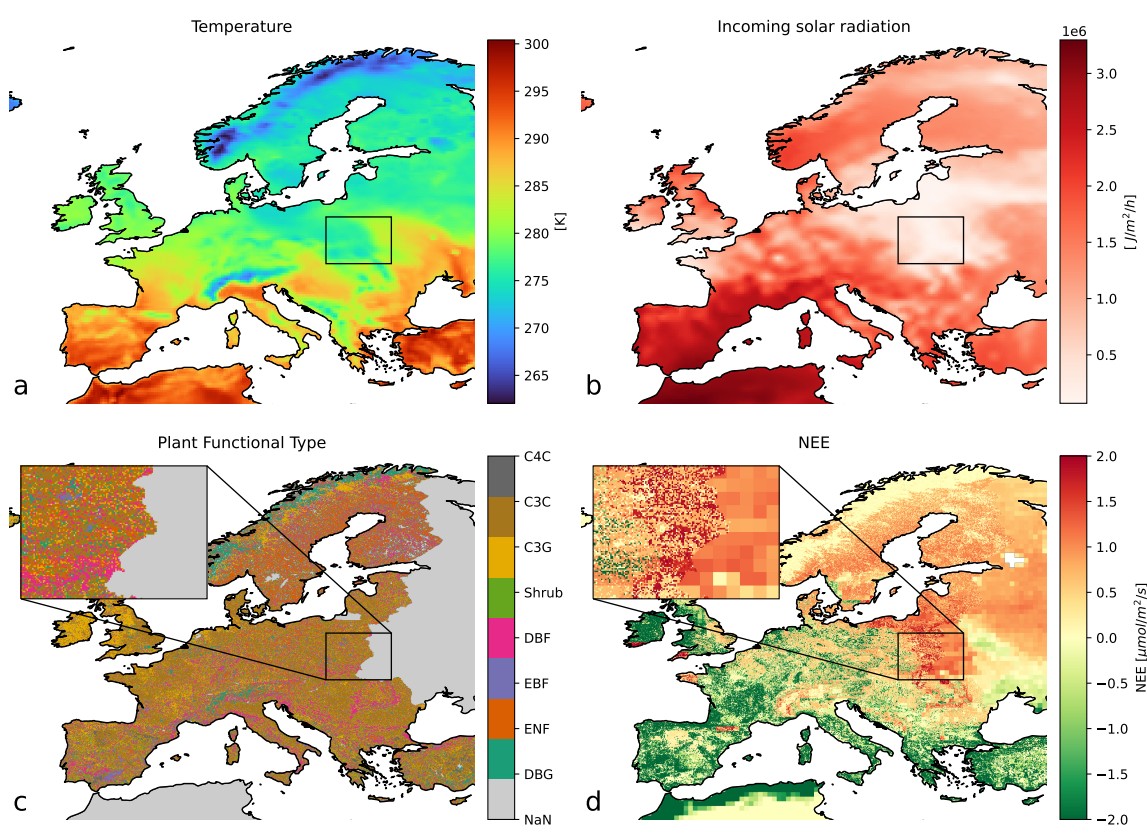

**Figure 3.** Temperature (a) and incoming solar radiation from ERA5 (b), high-resolution plant-functional type from the CORINE land-use classification (c) and NEE from CTE-HR over Europe (d) for April 1 2018, 14h UTC. The inset show a close-up on the border of where high resolution land-use data is available to illustrate the difference.

indicates a 10-year climatology over the last available years. Climatological ocean fluxes are created from the CarboScope product (Rödenbeck et al., 2013). $k$ is calculated following Wanninkhof (1992):

$$k = 0.31 * u^2 / \sqrt{Sc/660}, \tag{1}$$

where $Sc$ is the Schmidt number, calculated by a third-order function of sea surface temperature (Wanninkhof, 1992) and $u$ is the 10-meter wind speed. As $u$ and temperature are available on high-resolution, ocean fluxes can be calculated on a high-resolution as well. We note that more detailed European air-sea spatial flux patterns can be derived from ICOS observations of pCO2 (Becker et al., 2021), and a recently developed machine-learned monthly flux product can be considered to underlie our hourly fluxes if regularly updated.

## 2.2 Validation of the downscaling

### 2.2.1 SiB4 downscaling

A validation of the downscaling of the CTE-HR SiB4 fluxes is shown in Figure 4, where we show the spatial correlation coefficient between GPP fluxes in CTE-HR, and MODIS-derived Near-Infrared Reflection of vegetation (NIRv, see Badgley et al. (2019)), both at 0.05° (where CTE-HR fluxes were regridded using nearest-neighbour resampling). The correlation is calculated over N=100 high-resolution pixels within each of the larger 0.5 by 0.5° boxes for July 2016. A positive correlation coefficient between observed NIRv and simulated GPP suggests that credible sub-0.5 degree gradients were present in the high-resolution CTE-HR fluxes, even though they were originally calculated at a coarser (0.5°) resolution in SiB4. We find that for 1795 (56%) of the larger gridboxes, the spatial downscaling indeed represents the observed gradient in a statistically significant correlation. For 507 larger gridboxes (16%), the observed spatial gradient is misrepresented, and for 894 boxes (28%), no conclusions can be drawn due to lack of observed variability or lack of a significant correlation within the 0.5 by 0.5° gridbox. A similar result was found for other months in the growing season (when GPP is higher), and demonstrates that the downscaled biosphere fluxes are in the majority of boxes better than (54%), or at least as good as (28%), the SiB4 fluxes without downscaling.

### 2.2.2 Added value of the ENTSO-E data

The added value of ENTSO-E power usage data -relative to monthly data as is used widely in the community- mainly shows in specific cases where deviations from the mean flux are large. An example of this occurred during Christmas 2017 in Germany when, due to high wind speeds and an abundance of sustainable energy, German electricity prices went negative. This event was widely covered in media (e.g. Berke (2017)). With the ENTSO-E data, our CTE-HR emissions capture this increase in wind-generated electricity, and the corresponding decrease in energy generation by the combustion of fossil coal and gas. This is shown in Figure A1 in Appendix A, demonstrating the added value over a lower temporal resolution view such as provided by the CAMS emission dataset. We emphasize that the difference between the public power flux according to CAMS-REG-

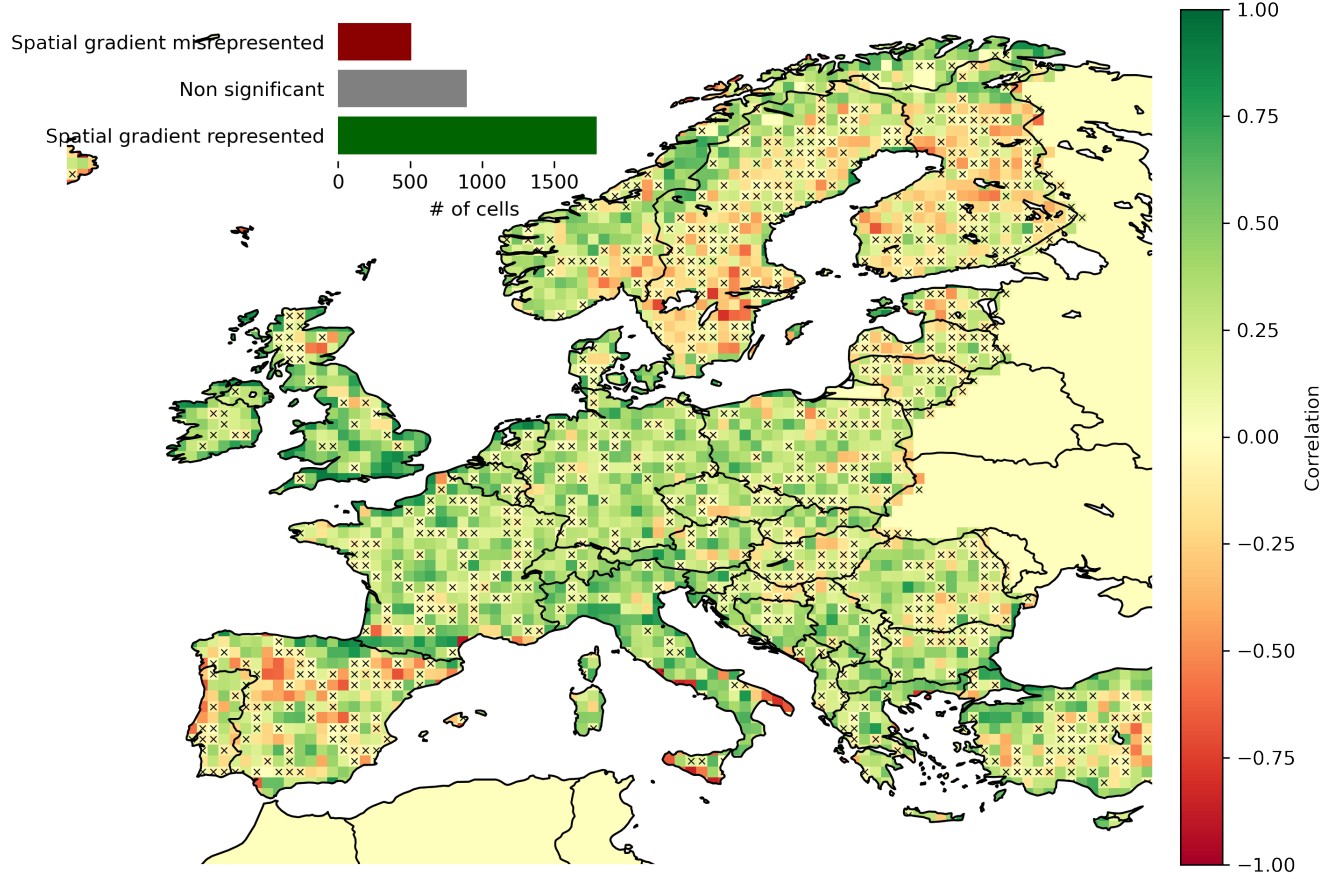

**Figure 4.** Correlation coefficient between NIRv and the downscaled SiB4 product from CTE-HR within 0.5x0.5 degree gridcells for July 2016. Non-significant values are marked with a cross. The inset in the upper left shows the amount of gridcells where the spatial gradient was represented, misrepresented, or that no significant correlation was found.

GHG emissions and the ENTSO-E data is about 1/3rd of the biosphere fluxes in Germany during December, and an error due to monthly constant anthropogenic emissions is thus unlikely to be corrected using atmospheric in-situ or space-based data. Instead, the reduced emissions from power generation would be wrongly attributed to the biosphere fluxes, or to other larger emissions sources if no sub-monthly data on power generation was available in the underlying emissions.

## 2.3 Methodology for the comparison to atmospheric measurements

### 2.3.1 Poor-person's inversion

We compare our high-resolution system (CTE-HR) to a poor-person's inversion (PPI), similar to the poor-mans inversion from Chevallier et al. (2009). The PPI is a relatively simple way of estimating the biosphere fluxes based on the global atmospheric

growth rate of $CO_2$ and used as a benchmark here. In our PPI, global $CO_2$ fluxes from anthropogenic emissions, ecosystem respiration, ocean exchange and wildfires are summed, and compared to the atmospheric growth rate of $CO_2$. Prior gross primary production is then scaled, so that the sum of the fluxes follows the global atmospheric growth rate:

$$\alpha GPP + TER + FF + Ocean + Fire = G, \tag{2}$$

where $GPP$ is the prior gross primary production, $\alpha$ is a scaling factor to close the budget, $TER$ is the total ecosystem respiration, $FF$ the anthropogenic $CO_2$ fluxes, including anthropogenic combustion emissions and cement production. $Ocean$ is the oceanic $CO_2$ exchange, $Fire$ are the $CO_2$ emissions by wildfire and $G$ is the monthly atmospheric growth rate of $CO_2$. We chose to scale GPP in the PPI, as directly scaling NEE resulted in non-physical fluxes (e.g. flipped diurnal cycles). Compared to TER, GPP has a larger inter-annual variability (Piao et al., 2020) and we therefore expected GPP to be a larger contributor to changes in the atmospheric carbon content than TER. We calculate TER and the prior GPP from a 10-year climatology of SiB4 (Haynes et al., 2019). This climatology was calculated over the 10 most recent years (2007-2017), based on a run that uses a spin-up iterated 5 times over 2000 to 2020. To get a correct representation of the carbon pools, we include fires in this spin-up. A diurnal cycle of GPP and TER is imposed, based on a SiB4 simulation with a constant atmospheric $CO_2$ of 370 ppm and no fires. The 3-hourly output was interpolated linearly to hourly values. The atmospheric growth rate $G$ is calculated from the NOAA global growth rate (https://gml.noaa.gov/ccgg/trends/gl_data.html) and a conversion factor of 2.086 GtC/ppm, similar to Chevallier et al. (2019). Anthropogenic $CO_2$ emissions are taken from GridFED from the previous year (Jones et al., 2021) and binned to a 1x1° grid. We do not extrapolate the anthropogenic $CO_2$ emissions, as these changes in emissions are generally quite small (within a few percent) and unknown in near real-time. Daily ocean emissions are taken from a climatology of the 10 most recent years from the CarboScope ocean inversion and bi-linearly interpolated to 1x1° (Rödenbeck et al., 2013). Fire emissions for the current month are taken from GFAS (Di Giuseppe et al., 2017) and binned to a 1x1° grid.

Note that our approach differs slightly from Chevallier et al. (2009), as they scale NEE based on the uncertainty in their biosphere model. For this, they use a static, manually fitted scaling factor, whereas we exactly follow the monthly growth rate.

### 2.3.2 Large-scale atmospheric transport

To compare the fluxes from CTE-HR to the PPI, we assess mole fraction residuals at the continental scale. Hitherto, we propagated the different fluxes (Sections 2.1 - 2.3.1) using the atmospheric transport model TM5 (Krol et al., 2005). We sampled $CO_2$ mole fractions at European ICOS sites (Ramonet et al., 2020; Drought 2018 Team and ICOS Atmosphere Thematic Centre, 2020). Similar to Smith et al. (2020), we use a global resolution of 3°x2° and nested zoom regions of 1x1° over the North America and Europe. Atmospheric transport is driven by meteorological fields from ERA-5 (Hersbach et al., 2020). Although the 1-degree resolution is coarser than the high-resolution fluxes, we here mostly show temporal variability in the $CO_2$ fluxes and mole fractions. Moreover, ICOS sites are generally located far away from large urban areas, allowing very local (FF) sources to mix through the atmosphere before they arrive at a measurement site, making high-resolution atmospheric transport less important. To get an overall idea of the performance of CTE-HR, we selected one site from each available country. In this

selection, we made sure that the selected sites cover a range of latitudes, longitudes and flux landscapes. The selected sites are shown in Figure 2. For these sites, we evaluate the root mean square error (RMSE) and correlation coefficient (R) between the simulated and observed $CO_2$ mole fractions. For the analysis with TM5, we discarded the 2.5% days with highest and lowest RMSE, to remove events that are not captured by the transport model.

As our product is not informed by large-scale constraints, we do not expect it to perform well over multi-annual timescales. However, as it contains much information on temporal variability, we do expect it to perform well on shorter, synoptic timescales. Therefore, we restart TM5 every month of 2018 from an initial $CO_2$ field taken from the CarbonTracker Europe (CTE2021) contribution to the GCP2021 release (Van der Laan-Luijkx et al., 2017; Friedlingstein et al., 2022a) and transport the fluxes for 5 weeks. Outside the high-resolution domain, we use PPI fluxes for atmospheric transport by TM5.

### 2.3.3 High-resolution transport

To test higher resolution transport, we analysed simulated and measured hourly mole fractions at the Lutjewad station in the Netherlands (53.24°N, 6.21°E) (LUT in Figure 2). In Lutjewad, fossil fuel emissions, biosphere exchange, and advection of background air (from the North Sea) shape the measured $CO_2$ record (Bozhinova, 2015; Van der Laan et al., 2010).

The mole fractions are the result of transport by the Lagrangian particle model STILT (Lin et al., 2003) at 0.1 by 0.2° resolution. The STILT model was driven by 3-hourly meteorological fields from the ECMWF-IFS short term forecasts (following the IFS cycle development, for more information see: https://www.ecmwf.int/en/publications/ifs-documentation. We released 100 particles and followed them for 10 days back in the atmosphere. We estimated the background $CO_2$ signal taking into account the average location of the 100 particles at the end of the 10-day back-trajectory. The STILT domain covered the same spatial extent as the CTE-HR domain. Background mole fractions were taken from CTE2021. We compare the high-resolution product CTE-HR to a 1x1° version of CTE-HR (coarse), and CTE-HR product without any temporal variability (flat, i.e. the temporal average of 10 previous days). For each hourly time interval, we selected the cases where the difference in mole fraction between the three versions of the CTE-HR model is larger than 2 ppm. These cases indicate relatively large differences between transport of the full resolution fluxes, and those with a lower spatial resolution, or a flat temporal profile. To analyse the influence of high-resolution transport on the capability to resolve different sectors, we transported each fossil fuel sector individually.

### 2.3.4 Local fluxes

To assess the performance of our model on a local scale, we compare the CTE-HR fluxes in Amsterdam to an eddy-covariance tower in the centre of the city. The tower is located at 52.366548°N, 4.893020°E at about 40m above ground level, about 20m above the average building height (Steeneveld et al., 2020). Note that the footprint of this tower (about 500m) is much smaller than a grid box in CTE-HR (about 15km). To be more representative of Amsterdam, we average the four grid boxes around the eddy-covariance tower.

## 3 Results

We performed several tests to assess the performance and limitations of our modelled fluxes. First of all, we assess the skill of CTE-HR to deal with anomalous events in the biosphere and for fossil fuel emissions. Secondly, we assess how well our fluxes can be used to represent the measured $CO_2$ mole fractions in the atmosphere over the entire European continent. We also performed a similar test on a much smaller scale with a higher resolution transport model to assess the benefits of the high spatial and temporal resolution of the CTE-HR fluxes. Finally, we compare CTE-HR fluxes with those of eddy-covariance $CO_2$ flux measurements in the city of Amsterdam to assess the representativeness of the diurnal cycle of our fossil fuel emissions in urban areas.

### 3.1 Continental and monthly scale: anomalies over Europe

Our fluxes are designed to be versatile enough to represent the biosphere and fossil fuel emissions in both normal and anomalous years. We illustrate this capability using two cases: the biosphere response to the 2018 European drought and the changes in fossil fuel emissions due to the 2020 COVID-19 restrictions.

Our CTE-HR biosphere flux anomalies during the 2018 European drought follow those of CTE presented by Smith et al. (2020), which were the result of inverse modeling of atmospheric $CO_2$ mole fractions (Figure 5). Similar to Smith et al. (2020), the CTE-HR fluxes show enhanced spring uptake in 2018 over Europe compared to 2016-2017, as well as reduced uptake during the summer drought (see Figure 5a and b, respectively). This progression is also shown in panel c, showing the total biosphere anomaly in the area influenced by the drought, as defined by Smith et al. (2020). Although both patterns in Figure 5c are similar, differences over the affected area of roughly 20 TgC/month are present for May-August. This shows that both the spring uptake and the drought response are underestimated in the SiB4 fluxes, which is very similar to the model set-up used for the prior estimate in Smith et al. (2020) (not shown). Atmospheric measurements hence added significant value to the prior SiB4 model in this case. Nevertheless, the similarity between the flux products indicates that we already capture anomalous periods in the biosphere reasonably well, without the need for computationally and time costly inverse modeling and delays due to data availability. CTE-HR therefore allows early recognition of such anomalies and possibly more rapid analyses of available atmospheric observations such as those collected by ICOS.

Global anthropogenic combustion emissions in 2020 decreased due to the global COVID-19 pandemic (Guevara et al., 2022; Le Quéré et al., 2020; Dou et al., 2021), something also visible in our European fossil fluxes (Figure 6). In CTE-HR, total European emissions decreased by 7% in 2020 compared to 2019, which is consistent with the values reported by carbonmonitor.org (Liu et al., 2020) (who report a decrease of 7.5%, not accounting for international aviation) and Guevara et al. (2022) (who report a decrease of 7.8% and 3.3% for fossil fuel and biofuel $CO_2$ emissions, respectively). Figure 6 shows that the decrease is highly sector-specific (lower panel), with the aviation sector showing the highest percent decrease (80%) during lock-downs (indicated in grey shading), compared to 2019. Also on-road, industry and shipping emissions decreased (30%, 30% and 20% maximum, respectively). As the on-road and industry sector contributed more to the total $CO_2$ emissions in the domain (see Figure 6, upper panel), their decrease impacts the total reduction more than the aviation sector. The found

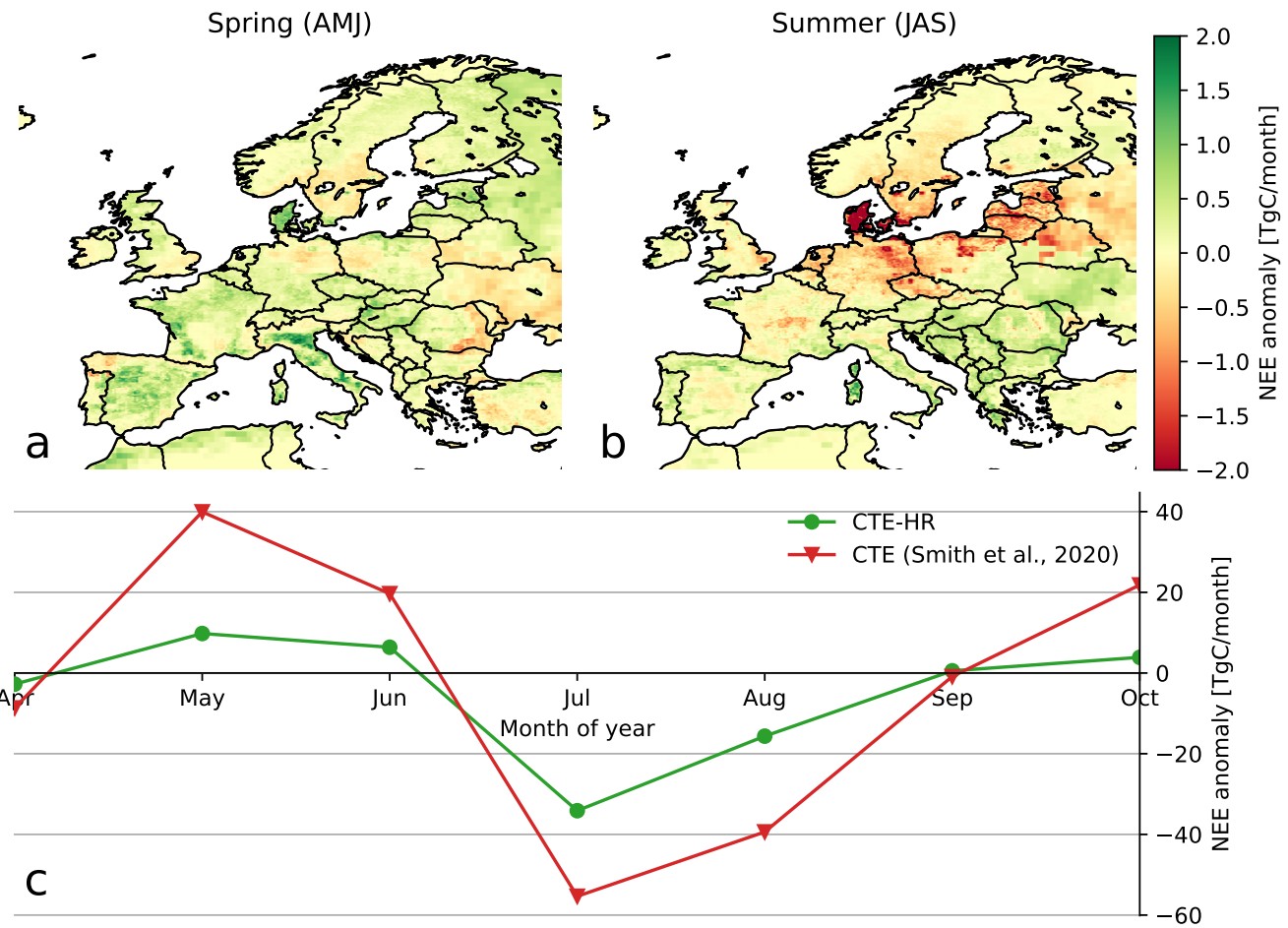

**Figure 5.** Estimated biosphere flux anomalies in 2018 compared to 2016-2017 for the spring (a) and summer (b), and the progression of the drought under the area influenced by the drought according to Smith et al. (2020) (c). Anomalies following Smith et al. (2020) are also shown for comparison. Note that Smith et al. (2020) show their anomalies relative to 2013-2017.

reductions are similar to Le Quéré et al. (2020) and carbonmonitor.org, who show a decrease of roughly 80% in the aviation sector, and 30% for the industry sector during lock-downs. Note that the emissions from the industry sector are estimated by Le Quéré et al. (2020) based on plants in the US and China. Contrary to Le Quéré et al. (2020), we do not find an increase in household heating emissions due to the COVID-19 confinement, but we note that in CTE-HR, household emissions only respond to temperature and not to socioeconomic changes. Overall though, the results presented here show that the fossil fuel emissions from CTE-HR respond to socioeconomic changes in a realistic way and hence capture much of the expected emission variability over time scales of weeks or months.

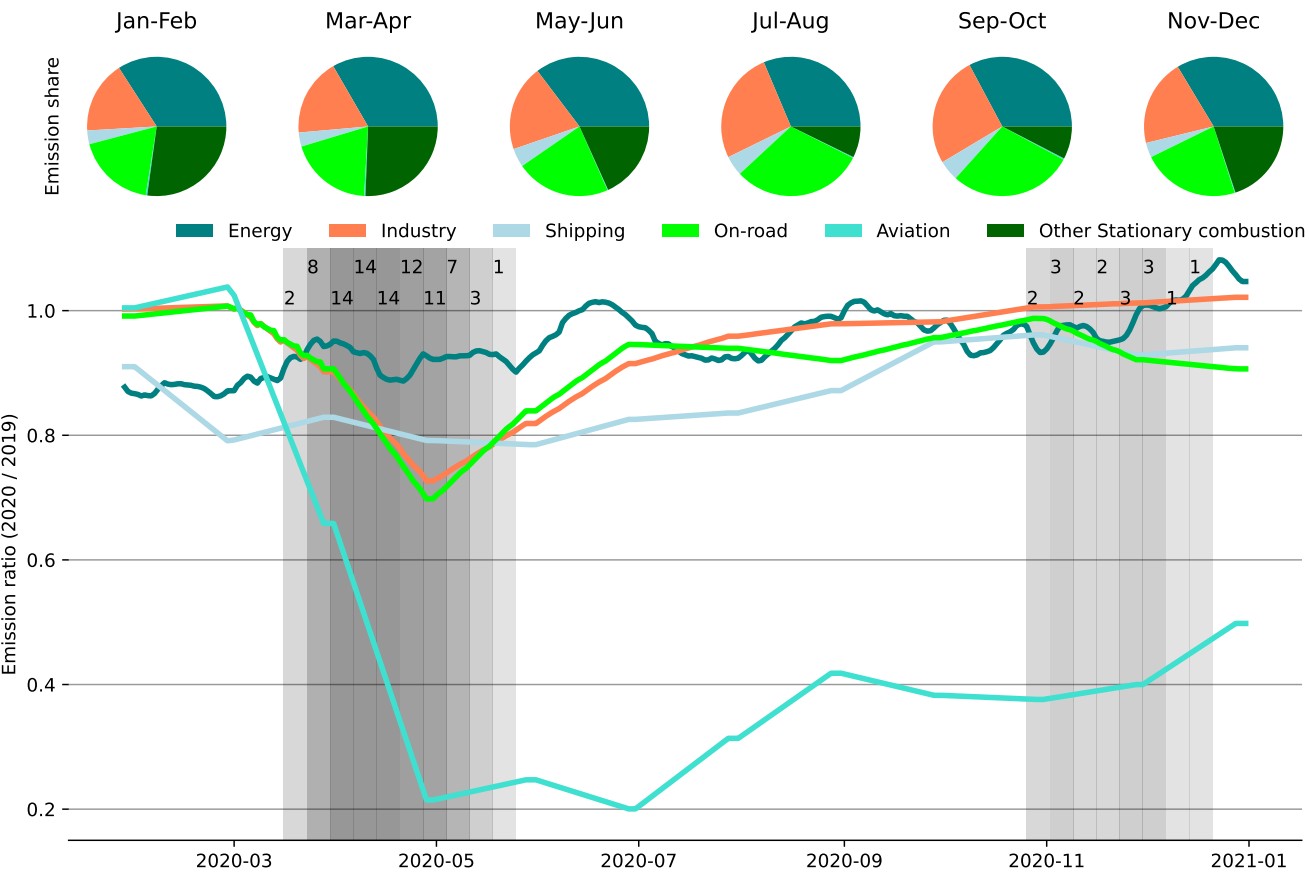

**Figure 6.** Relative sector share to total emissions in 2020 for two-month periods is shown in the pie-charts; Relative emissions compared to 2019 per sector calculated by a 28-day rolling average are shown in the line graph. The grey shading indicates periods with lock-downs, and the amount of European countries (out of 30) in lock-down as given by ECDC (2021) are indicated by the number in the grey bar. Household emissions are not included the lower panel, as household emissions only respond to temperature here. Note that aviaion has a relatively small share, as we only include emissions during landing and takeoff, following GNFR sector definitions.

**Table 6.** Fraction of the station months over all stations (N=46) for 2018 with a better statistical score than the threshold given in the header. The days with the highest and lowest 2.5% RMSE are discarded to remove events that are not captured by TM5, resulting in different amount of site-months for the different runs.

| | Threshold | | | | | |
|---|---|---|---|---|---|---|
| | Correlation [-] | | | RMSE [ppm] | | |
| | 0.9 | 0.7 | 0.5 | 2 | 4 | 6 |
| CTE2021 (N=433) | 0.05 | 0.44 | 0.66 | 0.12 | 0.56 | 0.84 |
| CTE-HR (N=451) | 0.04 | 0.48 | 0.76 | 0.08 | 0.57 | 0.85 |
| PPI (N=454) | 0.04 | 0.43 | 0.72 | 0.09 | 0.52 | 0.80 |

## 3.2 Continental and monthly scale: mole fractions

CTE-HR is designed to be a good first guess flux estimate for atmospheric modeling/data assimilation of $CO_2$. Hence, it will have added value if the transported fluxes result in simulated $CO_2$ mole fractions that are at least as close to the measurements as other methods that can be applied on a similarly short timescale. To test this, we compare our simulated mole fractions to a PPI (see Section 2.3.1), and mole fractions from CTE2021 contribution to the Global Carbon Project (Van der Laan-Luijkx et al., 2017; Friedlingstein et al., 2022a), as simulated by TM5 across a selection of European ICOS sites (see Figure 2).

On average, our transported fluxes result in better mole fractions (median RMSE=3.96) at European ICOS sites compared to the PPI inversion (median RMSE=4.32), rivalling those of the CTE2021 optimized fluxes (RMSE=3.95). This is indicated by the RMSE at the selected ICOS stations, indicated in Figure 7. For most stations we find a slightly higher RMSE compared to the inverse results of CTE2021 for the CTE-HR fluxes, but a lower RMSE than for PPI. This difference becomes more pronounced near high-emission regions where CTE-HR sometimes outperforms CTE2021 (e.g. at LUT, BIR, and BRM). Summarising this across all sites confirms this good performance in mole fractions across Europe, and we confirmed that these results are not sensitive to the choice of stations by assessing also the performance for selections of other stations. Corresponding to the lower RMSE, CTE-HR shows slightly higher correlations than CTE2021, as summarised for all station in Europe (N=46, see Drought 2018 Team and ICOS Atmosphere Thematic Centre (2020) with the exception of Zeppelin and Station Nord, that fall outside our domain) in Table 6. This suggests that additional temporal variability is resolved with the high spatio-temporal resolution of the underlying fluxes. Overall, the CTE-HR product scores better than the PPI, which confirms that the dynamical modeling through proxies, such as temperature, and sub-continental gradients added through SiB4 represent true flux variations which would not be captured by simply projecting a global $CO_2$ growth rate onto a climatological GPP map for Europe.

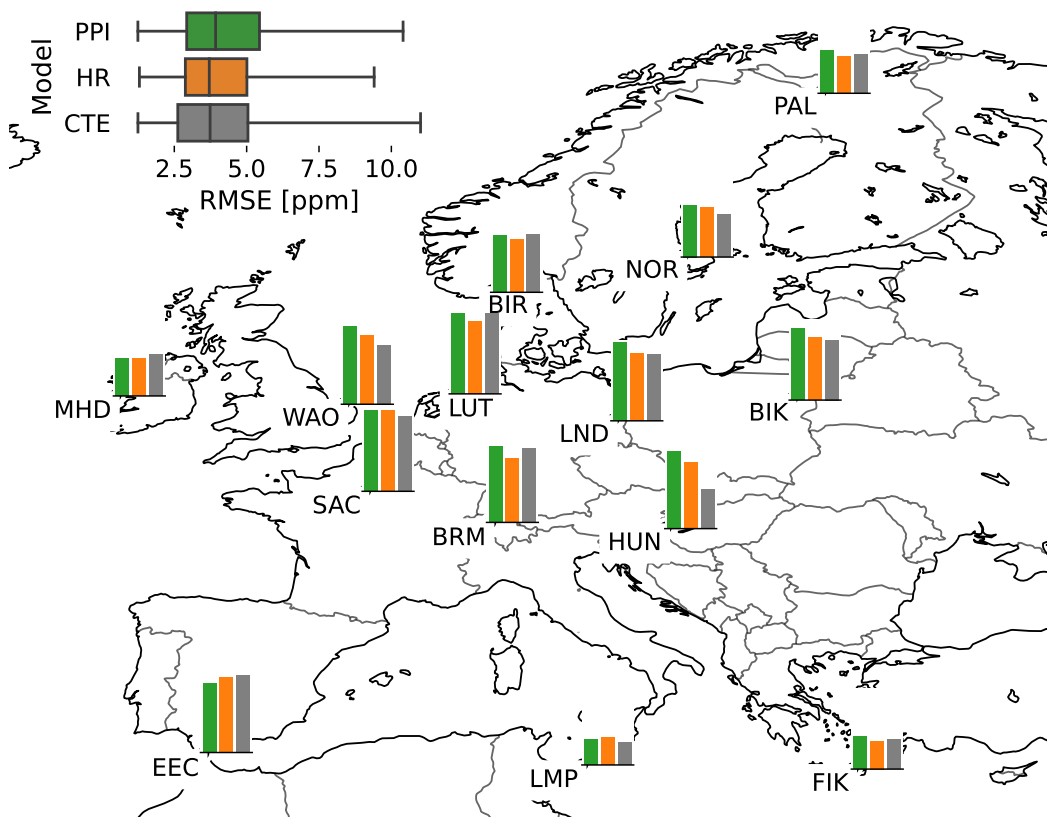

**Figure 7.** Root mean square error (RMSE) at selected stations in Europe. In the top left, a box plot of the monthly RMSE values is shown. PPI is the poor-person's inversion, HR is the CTE-HR flux (this work) and CTE2021 is the latest CarbonTracker Europe release. Note that all bars have the same y-axis, which has a maximum of 6 ppm.

### 3.3 Regional and daily scale: Lutjewad

CTE-HR outperforms the PPI at the European continental scale, but it is also designed for use in regional or country scale analyses. To assess the performance of our fluxes at the such scales, we analyse high-resolution transport at the regional scale for a selected site.

The time series of observed $CO_2$ mole fractions at the Lutjewad tower (Figure 8) are generally well reproduced by the CTE-HR flux estimates, when transported by STILT during well-mixed conditions (here assumed between 12:00 and 16:00 local time). During stable night-time conditions, CTE-HR underestimates the $CO_2$ mole fractions (see Figure 8b and c), which is a common problem in atmospheric transport modeling (Geels et al., 2007). Night-time and early-morning observations most strongly reflect this, especially in winter months, when fossil fuel plumes, along with respired $CO_2$ from surrounding agricultural fields, contribute significantly to the observed peaks in the $CO_2$ signal. In the night (22h - 4h local time), RMSE between the observed and simulated mole fractions is about 10 ppm, whereas during well-mixed conditions (10h-16h local time), the RMSE is about 6 ppm.

When we compare the transported fluxes of the CTE-HR product with a coarse (1x1°, low resolution) or temporally flat version of CTE-HR fluxes, we generally see only small differences (Figure 8a). Most notably, the largest differences between the high-resolution and temporally flat fluxes are when the biosphere is very active. In 64% of the simulated hours in May, this difference is larger than 2 ppm. This generally occurs during the night, when the boundary layer is shallow and respiration dominates (see Figure 8b). Of this 64%, 99.2% is dominated by differences in the prescribed biosphere fluxes. In contrast, the largest differences between the high-resolution and the spatially coarsened fluxes is in December (see Figure 8c), when the anthropogenic emissions are higher, and biosphere fluxes smaller. For 28% of the hours in December, the difference between the coarsened and high-resolution fluxes is larger than 2 ppm. Of these cases, 80% is dominated by differences in the prescribed Public power (GNFR A) fluxes. Note that the difference between the flattened and high-resolution fluxes in December is larger than 2 ppm in only 3.6% of the hours.

Overall, our transported high resolution fluxes result in good model performance for $CO_2$ at the Lutjewad tower. Differences between the high-resolution and the spatially coarsened (1x1°) fluxes are mainly seen in the 10-20% of the record when fossil fuel emissions are the dominant source of $CO_2$. Within the FF emissions, the largest differences are due to the public power sector, which has very local sources (power plants). The dominance of FF emissions here is not unexpected, as fossil fuel emissions vary orders of magnitude at regional spatial scales. Therefore, the main advantage we found for CTE-HR fluxes is that higher spatial resolution enables better resolving of emission plumes from point sources. We found similar advantages, and similar percentages, for high spatial resolution fluxes at the nearby Cabauw tower (not shown), where point sources such as from power plants affect the measured $CO_2$ in a few percent of the hourly data.

### 3.4 Local scale: Amsterdam urban fluxes

Eddy covariance measurements in Amsterdam as shown in Steeneveld et al. (2020) allow us to evaluate some aspects of our CTE-HR fossil fuel emissions in urban areas. Especially the short-term variability in the urban fluxes can be tested, since the

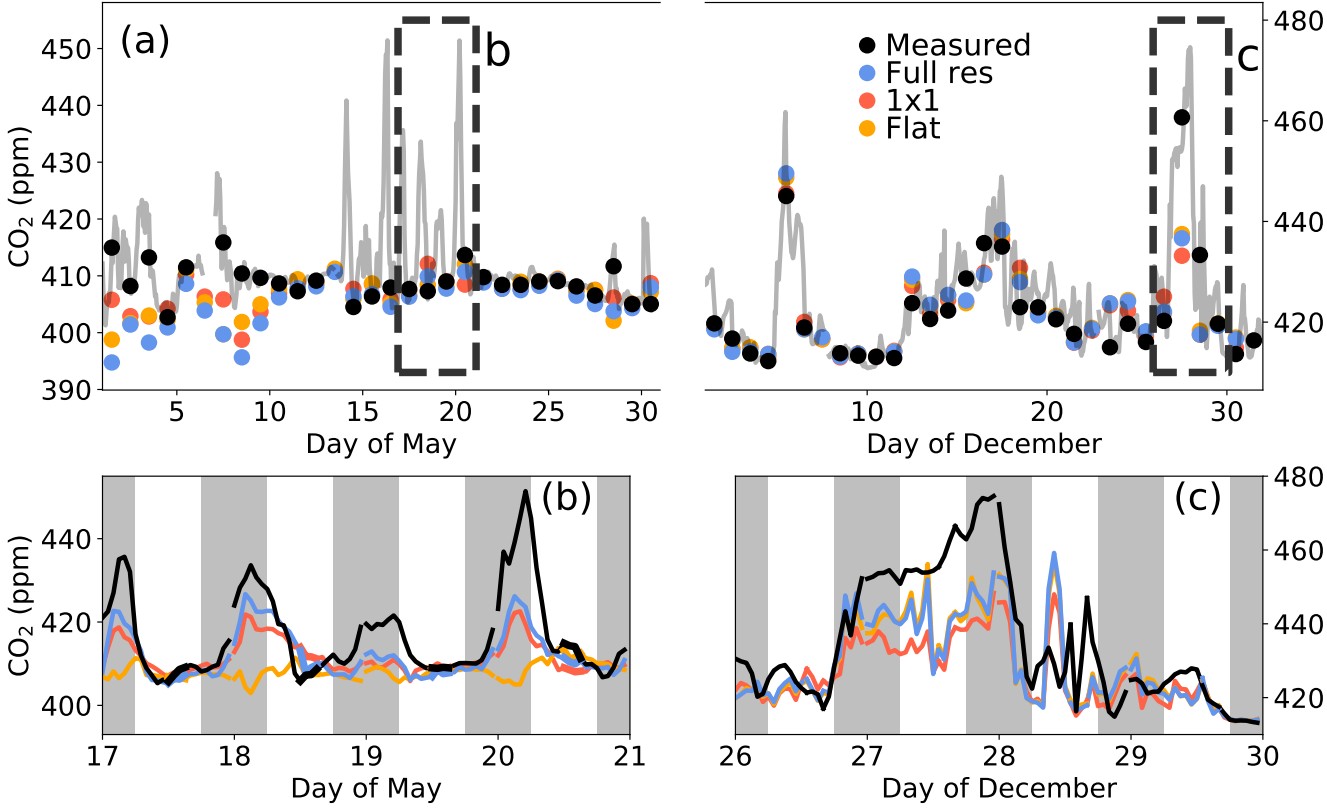

**Figure 8.** (a) Measured $CO_2$ time series at Lutjewad in the north of the Netherlands (53.24°N, 6.21°E) (grey lines), with a daily mean during well-mixed conditions (12:00-16:00 local time) indicated by black dots. Coloured dots indicate afternoon mean (12:00-16:00 local time) modeled mole fractions for STILT transport of our flux product at full resolution (blue), transport using a 1x1° coarsened version of our fluxes (red), and a version of our fluxes with flat diurnal cycles (orange). (b) and (c) show zooms of (a) for May and December respectively, with the full modelled time series. The periods between 18:00-06:00 are marked in grey to indicate the nights.

measurements more directly relate to the actual urban fluxes, and are not the result of an integrated signal over time as the $CO_2$ mole fractions.

Both the measured and the modeled fluxes show a distinct diurnal cycle (Figure 9), with maximum fluxes during the daytime and strong increases in flux in the morning. In summer, the peak of emissions is at midday, whereas in winter the peak of emissions occur later in the day, which is reasonably well captured by CTE-HR. The magnitude of the estimated fluxes is

lower than those in the eddy covariance measurements, especially in winter. However, since the footprint of the eddy covariance measurements cover a much smaller area (~500m) than our 0.1x0.2° grid cells (Steeneveld et al., 2020), the magnitude of the fluxes will also be affected by a difference in land cover within the footprint. For instance, part of the averaged grid cells of the CTE-HR emissions are covered in water, and an industrial area is located in one of the four grid cells. This makes direct

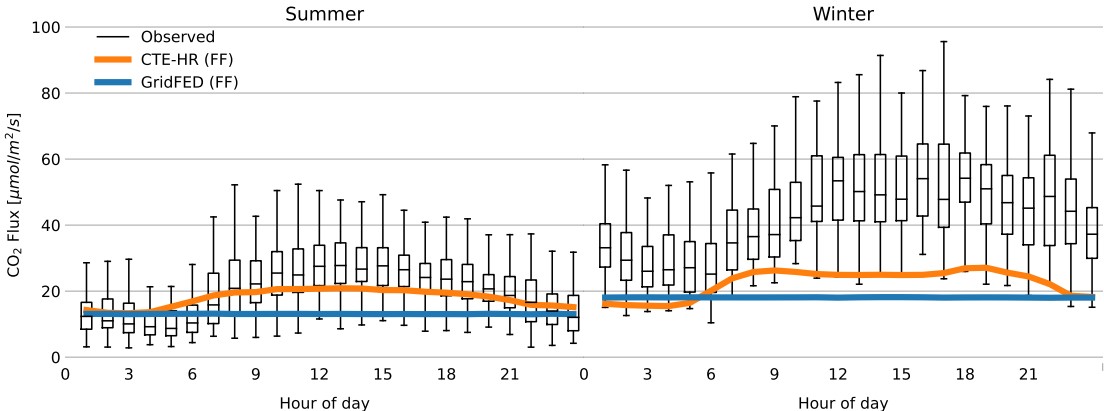

**Figure 9.** Average fluxes per hour of the day for summer (MJJA) and winter (ND) of 2018 over Amsterdam. The box plots show the observed fluxes. Orange shows the Amsterdam emissions as calculated by our CTE-HR model, averaged over 4 grid boxes over Amsterdam (52.3°N, 4.6°E to 52.5°N, 4.8°E). The blue line shows the GridFED emissions Jones et al. (2021), averaged over the same area as the CTE-HR emissions. September and October are not included in this figure, as there are large gaps in the observations in these months. Also the first months of the year are missing, as the measurements started in May 2018.

comparison of the magnitudes of the CTE-HR fluxes to the observations difficult. Despite these limitations of the comparison, it is clear that our diurnal and seasonal cycles add significant information compared to a flat profile (Figure 9).

## 4  Discussion

We present our high-resolution $CO_2$ CTE-HR flux product that provides European scale carbon fluxes 2 months after real-time with a 0.1° by 0.2° horizontal resolution. Below, we will discuss the anthropogenic and biosphere flux models, the atmospheric transport and a future outlook for CTE-HR. We end the discussion with our envisioned use of CTE-HR.

### 4.1  Anthropogenic emissions

On the Europe-wide annual scale, we find that our derived fossil fuel emission estimates agree well with state-of-the-art products such as GridFED (Version 2021.3) (Jones et al., 2021), showing the a similar trend (not shown). Note however, that GridFED provides only FF $CO_2$, whereas CTE-HR also includes biofuel emissions.

For the COVID-19 period in early 2020, we find a decrease in $CO_2$ emissions from industry, aviation and ground transport and residential heating. These reductions mostly correspond to the findings of Liu et al. (2020) and Le Quéré et al. (2020), although the latter did not find reductions in the residential sector. We attribute this discrepancy to a difference in approach and study area, as Le Quéré et al. (2020) derived emissions for this sector from smart meters in the UK. Our approach, based only on temperature, results in reduced residential heating over Europe due to the warm winter, which was also shown by Liu

et al. (2020). Both approaches are complementary, and in principle the smart meter approach is expected to give more accurate
estimates, but these are not available Europe-wide, nor for all social-economic classes. On the other hand, our approach, based on temperature, is available for all residential areas. For energy production, both Liu et al. (2020) and Le Quéré et al. (2020) found a median decrease in $CO_2$ emissions from public power of roughly 10% in 2020 compared to 2019, which is similar to the median decrease of 11% that we find with CTE-HR. We do not find the regional increases in emissions over Europe for parts of France, Spain, Italy and Germany that Dou et al. (2021) obtained. We attribute these differences to differences and
changes in the spatial distribution of anthropogenic activity during the COVID-19 crisis. Dou et al. (2021) use satellite proxies to adjust regional estimates of emissions, whereas we use the most recent emission inventory data (Kuenen et al., 2022), which does not necessarily capture recent changes in the spatial distribution of emissions. The importance of such changes becomes evident during global crises, such as the COVID-19 pandemic, but can also be due to political and societal choices, such as expected decisions to move away from gas-use across Europe. Other spatial discrepancies are introduced by the public power
sector, which is the main fossil fuel sector. We use ENTSO-E data to resolve the temporal variability in this sector. However, we do not distinguish different types of power plants (with the exception of nuclear), such as gas or coal spatially. We do currently not use this in our CTE-HR system, as detailed information about the spatial distributions of anthropogenic emissions (such as population density and industrial area) becomes available after roughly 2 year (Kuenen et al., 2022). Note that these outdated spatial distributions contribute significantly to the total uncertainty on grid-cell level.

Other major sources of uncertainty in the anthropogenic emissions from CTE-HR stem from 1) the use of proxies, such as Eurostat economic indicators for the $CO_2$ emissions from the industry, 2) the carbon intensities of fuels, to translate energy generated to $CO_2$ emissions from public power; 3) temporal downscaling of yearly to hourly fluxes. An exact uncertainty estimate of the combined uncertainty in these sources is nearly impossible, as one has to account for all spatio-temporal correlations in the uncertainty structure. Our best estimate of the uncertainty in our anthropogenic fluxes is based on a similar
approach by Super et al. (2020a); Liu et al. (2020). Liu et al. (2020) found a daily, country-total uncertainty of 7%, using a similar methodology. Super et al. (2020a) suggested that the scaling of country total emissions (uncertainty of $\pm2\%$) down to the gridcell-level (1x1 km) increased the yearly uncertainty to 18% of the flux, assuming a Gaussian error distribution. If we assume the spatial scaling error to our 15x15km grid to also be 18% (a possibly somewhat high estimate), and for this error to be independent of the temporal scaling error of 7%, the addition in quadrature of these errors yields a total daily gridcell
uncertainty of 19.3% of the calculated anthropogenic $CO_2$ flux. Due to lack of knowledge about the correlation structure in these uncertainties, this is currently the best uncertainty estimate we can provide for the anthropogenic fluxes. Note however that some sectors (e.g. public power), have smaller uncertainties associated to them, and that therefore, generally, gridcells with larger fluxes have smaller uncertainties (Super et al., 2020a). The weight factors used in CTE-HR that represent diurnal and seasonal profiles of the anthropogenic $CO_2$ emissions do not include uncertainty estimates (Guevara et al., 2021), and
therefore we cannot provide an exact uncertainty estimate for the hourly fluxes. From Super et al. (2020a), we estimate an added uncertainty of 2% on country-total $CO_2$ fluxes, due to the temporal downscaling, looking only at well-mixed conditions.

## 4.2 Biosphere fluxes

### 4.2.1 Spatial downscaling

For CTE-HR, we applied further downscaling of our SiB4 fluxes using the CORINE land-use classes. The downscaling to 0.1°
by 0.2° using CORINE is based on the assumption that at the original resolution of SiB4 of 0.5°, differences in land-use are
more important for biosphere carbon exchange than meteorological variability, which we deem true for the synoptic time-scale
(see also Figure 3). However, the downscaling also depends on the translation from land-use class to plant functional type,
which is not straightforward for all land-use classes. An example of this is the land-use class 'arable land', which we translated
to C3 general plants. Nevertheless, resulting differences between the original SiB4 and the high-resolution biosphere fluxes
are small (<5% difference in total monthly flux for 2017-2021), and we consider the gain in resolution to outweigh any added
uncertainties.

### 4.2.2 SiB4 performance

On the European scale, the SiB4 biosphere $CO_2$ fluxes have previously been compared to Eddy covariance flux observations by
e.g. Smith et al. (2020); Kooijmans et al. (2021), and show a good comparison (RMSE of roughly 2 $\mu mol/m^2/s$ (Haynes et al.,
2019)). We also compared the PFT-aggregated mean of the net ecosystem productivity (NEP) from both FLUXCOM and Zeng
et al. (2020) to our CTE-HR product for the growing season (MJJA) of 2017. We find that both FLUXCOM and the product
by Zeng et al. (2020) have a higher NEP than CTE-HR but that the latter might better agree with regional integrals. Table 7
shows the differences, where the high NEP corresponds to earlier reports (Jung et al., 2020; Zeng et al., 2020) of large NEP
(globally integrated near 10 PgC/yr sink in FLUXCOM). It also agrees with a tendency for EC-based analyses to represent
high uptake locations rather than lower or average locations, leading to potential overestimates of the machine-learning derived
fluxes (Jung et al., 2020; Zeng et al., 2020). In contrast, fluxes optimized using data assimilation of atmospheric $CO_2$ mole
fraction observations from among others the European ICOS network in CTE Friedlingstein et al. (2022b) suggest a lower
NEP, with the integral matched by CTE-HR more closely than the other products for most PFTs.

Our CTE-HR fluxes show generally a lower GPP compared to FLUXCOM (which is arguably gives more reliable GPP than
NEP estimates), especially for needleleaf ecosystems found in Scandinavia and for C3 crops. The latter is a generic PFT in
SiB4, and mostly used as placeholder for specific crop species that are part of SiBCrop (Lokupitiya et al., 2009). The agreement
with FLUXCOM is generally better than with Zeng et al. (2020), and CTE-HR typically is slightly low on GPP. Additionally,
we compared biosphere fluxes of CTE-HR to eddy covariance flux observations, to assess the mean error (ME), and RMSE
and the correlation coefficient (R) (Appendix B). The differences for both NEP and GPP are generally within the "local" error.

## 4.3 Atmospheric transport

Our transported CTE-HR fluxes show a better agreement to $CO_2$ mole fraction observations compared to a poor-person's
inversion, which is a relatively simple way of generating near real-time flux estimates. In this comparison to observations, we

|  | ENF | DNF | EBF | DBF | SHRUB | C3GRASS | C3CROPS | C4CROPS |
|---|---|---|---|---|---|---|---|---|
| *GPP* | | | | | | | | |
| HR | 6.40 | 19.98 | 1.24 | 26.44 | 6.57 | 22.28 | 39.52 | 0.47 |
| FLUXCOM | 9.60 | 20.40 | 2.06 | 23.43 | 7.17 | 21.23 | 42.99 | 0.92 |
| Zeng | 11.36 | 24.92 | 2.68 | 31.58 | 9.64 | 25.62 | 52.49 | 1.21 |
| *NEP* | | | | | | | | |
| HR | -1.52 | -4.30 | -0.21 | -5.37 | -1.34 | -4.77 | -10.55 | -0.09 |
| FLUXCOM | -3.04 | -6.19 | -0.58 | -8.02 | -2.13 | -6.87 | -14.28 | -0.19 |
| Zeng | -3.23 | -7.56 | -0.73 | -10.45 | -2.95 | -7.42 | -15.60 | -0.29 |
| CTE | -2.48 | -4.09 | -0.76 | -6.32 | -1.94 | -5.44 | -14.12 | -0.29 |

**Table 7.** Gross primary production (GPP) and Net Ecosystem Productivity (NEP), integrated over the growing season (MJJA) of 2017 in TgC/month for different land-use types (ENF: evergreen needleleaf forests; DNF: deciduous needleleaf forests; EBF: evergreen broadleaf forests; DBF: deciduous broadleaf forests; SHRUB: Shrublands; C3GRASS: C3 grasslands; C3ROPS: C3 croplands; C4CROPS: C4 croplands. The land-use types are taken from the CORINE dataset (Bossard et al., 2000). HR refers to the product described here, Zeng to the product as described by (Zeng et al., 2020), FLUXCOM refers to the product by (Jung et al., 2020), and CTE refers to NEP optimized using data assimilation of atmospheric $CO_2$ mole fraction observations Friedlingstein et al. (2022b)

used both the relatively coarse resolution transport model TM5 (Krol et al., 2005) at 1° by 1° for Europe, as well as the high-resolution transport model STILT (Gerbig et al., 2003), driven by IFS meteorological fields, at 0.1° by 0.2° for the Lutjewad tower in the Netherlands specifically. Using the high-resolution fluxes we capture more variability compared to fluxes that are averaged to 1x1° and fluxes that have no temporal profile. Moreover, the high resolution allows us to study the effect of individual sectors, highlighting emission hotspots, such as power plants, as a category that benefits most directly from high-resolution fluxes and high-resolution transport. As we found similar results for the Cabauw tower in the Netherlands (not shown), we speculate that also for atmospheric $CO_2$ modeling at other European locations, the added resolution of the CTE-HR product will matter most for the specific wind directions and times of day when point sources contribute to the signal. Note that we assumed all emissions to be on the surface, which might bias stack emissions (Maier et al., 2022). Nevertheless, we do not expect this to influence the results of our comparison, as we assume this for the high-resolution fluxes, the 1x1° fluxes and the flattened fluxes. The largest fraction of observed $CO_2$ variability however is driven by synoptic variations and biospheric fluxes, even in an emission-dense region in the Netherlands where we assessed the Lutjewad and Cabauw tower records. As a result, the use of high-resolution fluxes (or low-resolution transport, not shown) does not directly affect our skill in simulating atmospheric mole fractions. This indicates that atmospheric transport models should be improved at the sub-synoptic time scales to study high-resolution fluxes in more detail, for example in their representation of the mixed layer height, as Lagrangian transport models are known to be sensitive to this.

The importance of atmospheric transport is also relevant for our analysis of the Amsterdam fluxes. We underestimate the fluxes, which we attribute to the small footprint of the flux tower, which is much smaller than the grid cells in our model

(Nicolini et al., 2022). A minor contribution to the underestimation is that we do not account for human respiration in our model, which attributes roughly 3% of the total $CO_2$ flux in urban areas (Ciais et al., 2020). However, our under-estimation is larger than the possible effect of human respiration. Additionally, we do not account for biosphere fluxes in Amsterdam, as we only compare to anthropogenic fluxes. In winter, biosphere fluxes are a source, resulting in higher $CO_2$ emissions. Contrary, in summer, the biosphere acts as a sink, offsetting the positive anthropogenic flux. Thereby, the biosphere can explain part of the smaller underestimation of the Amsterdam fluxes in summer, compared to the winter. Although our simulated fluxes are lower than the observed fluxes, we find a very good correlation between simulated and observed diurnal cycles (R=0.94), indicating that we capture the time profile of emissions in Amsterdam well. To better capture the absolute fluxes as seen by the flux tower, we should create higher resolution (<500m) fluxes, similar to the footprint of the flux tower. Although this is possible, it would be computationally expensive and we deem the 0.1 by 0.2° high enough to use as ready-to-use alternative for current European regional fluxes, especially given the limitations by current state-of-the-art transport models in urban environments.

## 4.4 Future outlook

Currently, CTE-HR provides biogenic, anthropogenic, ocean and wildfire $CO_2$ fluxes. However, for $CO_2$ emission verification, also other tracers and isotopes can be used (Balsamo et al., 2021) such as CO and $NO_2$ that are co-emitted with $CO_2$. CO and $NO_2$ column abundances can be monitored with satellites, and have been used for monitoring and verification of high-resolution anthropogenic $CO_2$ fluxes (Konovalov et al., 2016; Reuter et al., 2019). For these co-emitted species, a differentiation between different power plants should be included, as different fuels have different emission ratios of $CO_2$, CO and $NO_x$. For further improved MVS systems, biosphere fluxes should be disentangled from anthropogenic $CO_2$ fluxes. For this the radioactive isotope radiocarbon ($^{14}$C) can be used (Levin et al., 2011; Miller et al., 2020; Basu et al., 2020). However, $^{14}$C samples have to be analysed in a laboratory and therefore cannot be measured continuously, nor in near real-time (Levin et al., 2020). Oxygen ($O_2$) on the other hand, does not have this drawback. Oxygen is exchanged during different plant processes and consumed in the combustion of fossil fuels. By assessing $O_2$:$CO_2$ ratios, oxygen has previously been used to study the carbon budget of deciduous forests in the USA and Japan (Battle et al., 2019; Ishidoya et al., 2015), and to study the reduction of anthropogenic $CO_2$ emissions in the UK during the COVID-19 lockdown (Pickers et al., 2022). Furthermore, the isotopic signature $\Delta^{17}O$ in $CO_2$ is suggested as tracer for gross primary production (Hoag, 2005; Koren et al., 2019). and this tracer can also inform on the fossil fuel contribution to $CO_2$ mole fractions (Laskar et al., 2016). Measurements at the Lutjewad site are currently ongoing (Steur et al., 2021). To enable improved constraints on the European carbon budget, we aim to include estimates of the previously mentioned gases and isotopes in future releases. Measurements of these gases and satellite retrievals are generally available with a small latency (roughly 1 day, with the exception of the isotopes and oxygen) (e.g. https://doi.org/10.18160/ATM_NRT_CO2_CH4), and can therefore be used for a near real-time application as well.

With atmospheric $CO_2$ measurements being available within a few days, one might expect our flux product to have a similar latency. Currently, our latency of roughly 8 weeks is dominated by Eurostat statistical data and ERA5 meteorological data. Also other flux products such as carbonmonitor.org and GRACED (Dou et al., 2021) have this limitation. In a future update we aim to create a more near real-time emission estimate. With ENTSO-E data and ERA5 meteorological fields, we already

have near real-time information on public power and household emissions, as well as biosphere and oceanic fluxes. For a more complete budget, near real-time scaling of the other major sectors on-road and industrial should be included. Using near real-time atmospheric data, also atmospheric transport can be made near real-time. Operational transport of the near real-time fluxes gives a continuous verification of our European carbon fluxes and we aim to do this in a future release of this product.

## 4.5 Potential usage of CTE-HR

In contrast to other currently available near real-time, high-resolution flux products, our fluxes are designed to be used as an easy substitute for less-informed or lower resolution carbon flux products over Europe in modeling studies. CTE-HR is developed with the emphasis on estimating fossil fuel emissions and biosphere exchange rapidly, using information from emission proxies to estimate the recent state of European carbon exchange. Having noted this, it is not intended to be used as a policy tool directly, and generated fluxes are not a substitute for emissions reported by national emission registration entities.

## 5  Conclusions

We demonstrate and validate our new framework for estimating high-resolution carbon fluxes over Europe: CTE-HR. Its fluxes are created with a latency of about 8 weeks, and we show here that they can readily be used in atmospheric (inverse) modeling frameworks. The $CO_2$ fluxes provided by CTE-HR are driven by information on socioeconomic activity and meteorological data as dynamical proxies for variability that is unresolved in static emission inventories. We show that our fluxes reflect recent anomalies in both the European biosphere and economic activity due to the 2018 drought and COVID-19 lockdowns well, and after atmospheric transport they result in satisfactory agreement to $CO_2$ observations at European measurement towers at the continental scale. Individual emission sectors are resolved at high-resolution and can be separated in $CO_2$ mole fraction signals when transported to the Lutjewad tower in the Netherlands. The benefits of the high-resolution aspect of our CTE-HR fluxes are highest for the 5-10% of observed signals that are dominated by point sources, mostly from energy production. At even smaller scales, our fluxes represent the temporal variations well, but our estimated flux magnitudes are too coarse to be used for urban-scale carbon flux studies.

The CTE-HR system is built into the CarbonTracker Data Assimilation Shell (CTDAS) system (Van der Laan-Luijkx et al., 2017), allowing flexibility and potential use in inverse modeling studies. Future developments include the addition of other species, reduced latency, improved representation of biosphere fluxes and (automated) transport of the fluxes through the atmosphere to have an operational, continuous comparison to atmospheric mole fractions. The CTE-HR flux products are available on the ICOS Carbon Portal (https://doi.org/10.18160/20Z1-AYJ2), and we plan regular updates to stay within 2 months of real-time.

## 6 Code availability

The used code is available at https://doi.org/10.5281/zenodo.6477331 and a living repository can be found at https://git.
wageningenur.nl/ctdas/CTDAS/-/tree/near-real-time

## 7 Data availability

Fluxes generated by CTE-HR are available on the ICOS carbon portal https://doi.org/10.18160/20Z1-AYJ2. The fluxes contain modified Copernicus Atmosphere Monitoring Service Information [2022]. Neither the European Commission nor ECMWF is responsible for any use that may be made of the information it contains.

## Appendix A: ENTSO-E data

Figure A1 shows a specific case where the added information from the ENTSO-E data is valuable.

## Appendix B: Uncertainty of the biosphere model

We compared the CTE-HR biosphere flux to the observed NEP at 16 flux towers, where the land-use class is similar to the plant functional type that we use for the downscaling. We analyse the daytime mean fluxes (between 11h and 16h local time). We assess the Mean Error (the bias), the root mean square error (RMSE) and the correlation coefficient (R). As these metrics vary over the year, we denote them per season. The 16 used towers are: 'SE-Htm', 'BE-Bra', 'FI-Hyy', 'DK-Vng', 'DE-RuS', 'SE-Svb', 'FR-Bil', 'DE-Tha', 'BE-Vie', 'FR-Fon', 'SE-Nor', 'CH-Dav', 'DE-Geb', 'DE-HoH', 'BE-Lon', 'FR-Lam', 'IT-SR2' (Arriga et al., 2022; Brut et al., 2022; Heinesch et al., 2022; Rebmann et al., 2022; Bruemmer et al., 2022; Buchmann et al., 2022; Mölder et al., 2022; Dufrêne et al., 2022; Vincke et al., 2022; Bernhofer et al., 2022; Loustau et al., 2022; Peichl et al., 2022; Schmidt et al., 2022; Friborg et al., 2022; Mammarella et al., 2022; Janssens et al., 2022; Heliasz et al., 2022)

We compare our flux estimates at a specific grid cell to flux towers, which shows the expected mismatch when comparing our biosphere fluxes to eddy-covariance towers. As the used metrics vary over the year, we denote them per season in Table B1.

Note that we compare our 0.1° by 0.2° gridcells here to often pristine eddy-covariance sites.

*Author contributions.* AW, WP, ITL and RK designed the study, interpreted the results and wrote the manuscript, together with input from all other authors. AW, WP, RK and NS developed CTE-HR, with contributions from LK (SiB4), GK (TM5 simulations), ITL (CTE). SB provided the STILT footprints. GS provided the Amsterdam flux measurements, HAJM and BAS provided the Lutjewad data. InS created the initial model and with IdS contributed to the analysis of the fossil fuel module.

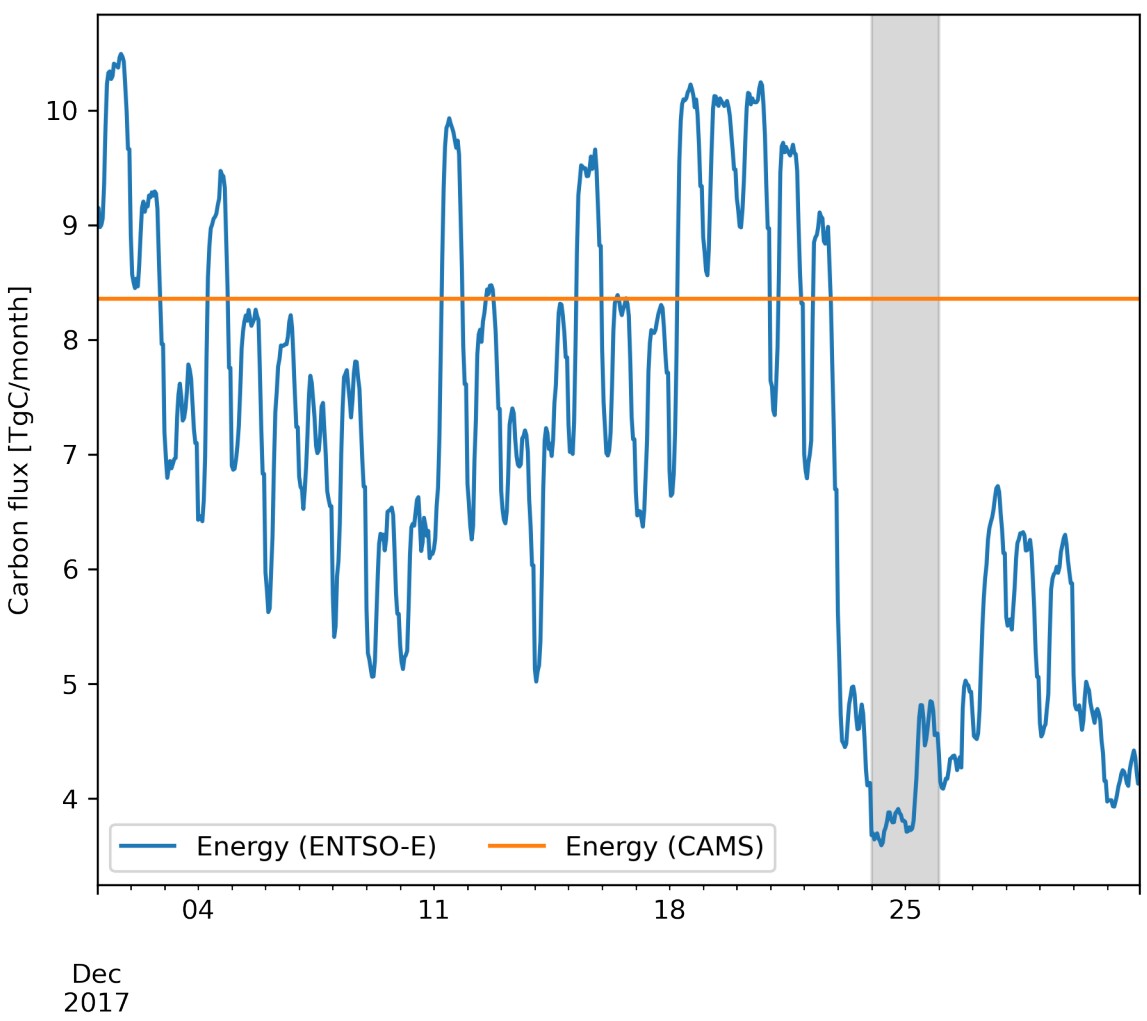

**Figure A1.** CO$_2$ emission from the energy sector, according to the ENTSO-E reported data and CTE-HR (blue line) and the CAMS-REG-GHG dataset (green line) over Germany for December 2017. The grey area indicates the period in which the negative energy prices occured is indictated

| Land-use type | Winter (DJF) | | | Spring (MAM) | | | Summer (JJA) | | | Autumn (SON) | | |
|---|---|---|---|---|---|---|---|---|---|---|---|---|
| | ME | RMSE | R | ME | RMSE | R | ME | RMSE | R | ME | RMSE | R |
| Evergreen Needleleaf Forests (N=10) | -1.33 | 2.49 | 0.54 | -5.24 | 6.20 | 0.70 | -4.49 | 5.63 | 0.42 | -3.81 | 4.87 | 0.74 |
| Croplands (N=5) | -1.70 | 2.58 | 0.33 | -7.75 | 15.10 | 0.48 | -7.10 | 15.06 | 0.08 | -4.02 | 5.36 | 0.19 |
| Deciduous Broadleaf Forests (N=2) | -0.07 | 1.48 | -0.07 | -3.15 | 6.76 | 0.81 | -12.28 | 13.10 | 0.50 | -4.97 | 7.26 | 0.79 |

**Table B1.** Mean error (ME, $\mu mol/m^2/s$), root mean square error (RMSE, $\mu mol/m^2/s$) and Pearson's correlation coefficient (R) of daytime means (11-16h) of the CTE-HR biosphere fluxes, compared to flux towers (N=17) at similar land-use types, for different seasons. The number provided is the median of the N sites per PFT (in parentheses)

*Competing interests.* The authors declare that they have no conflict of interest.

*Acknowledgements.* The authors acknowledge funding from the Netherlands Organisation for Scientific Research (NWO) Project 864.14.007. The observations of the Amsterdam Atmospheric Monitoring Supersite have been financially supported by the Amsterdam Institute for Advanced Metropolitan Solutions (AMS-Institute) under project VIR16002.

The authors acknowledge Christian Rödenbeck for providing the prior ocean $CO_2$ fluxes.

The authors acknowledge Michel Ramonet for the use of the Saclay measurements (SAC), and László Haszpra (HUN observations) with

funding by Hungarian National Research, Development and Innovation Office (grant no. OTKA K141839).

LK is funded through the ERC advanced funding scheme (COS-OCS, 742798)

The authors thank Zois Zogopoulos for facilitating the data provision on the ICOS Carbon Portal.

We thank SURF (www.surf.nl) for the support in using the National Supercomputer Snellius (NWO-2021.010/L1).

The authors thank two anonymous reviewers for their comments, which helped to improve the manuscript.

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
