# Peer review of "Near real-time CO2 fluxes from CarbonTracker Europe for high resolution atmospheric modeling"

_Earth System Science Data, 2022_

## Author Response (AR1)

**1 Reviewer 1 (Anonymous)**

**1.1 General comments.**

- *The manuscript presents a construction of surface CO2 flux dataset for European domain, designed for near-realtime updates and based on recent energy statistics and weather data. The flux dataset was evaluated via comparing the transport model-simulated timeseries to atmospheric observations and comparing fluxes to an urban flux tower data and demonstrated a reasonably good performance. The manuscript only requires minor revisions and can be accepted after implementing corrections and amendments based on suggestions provided below.*

We thank the anonymous reviewer for their positive, critical, and constructive feedback, which helped to improve the paper. Below, we address the reviewer's suggestions one by one, with the reviewer comments in italic.

**1.2 Detailed comments.**

- *It is advisable to enhance the part on validation of the SiB4 CO2 terrestrial biospheric fluxes (in terms as midday fluxes or daytime mean, where fluxes are more curtain) against data-driven products available at similar resolution ( 0.1): eg FLUXCOMM (Jung et al. 2020), Zeng et al (2020) or others, which can be reported summarily by region or dominant vegetation type. It is mentioned that Smith et al. 2020 did evaluation, but they did not have advantage of using a downscaled product which makes matching resolutions easier.*

We agree with the reviewer that the SiB4 validation could be enhanced. We see three possibilities for evaluation of SiB4: (1) local validation with e.g. FLUXNET data, (2) integrated validation with atmospheric data, and (3) a regional comparison to existing products, as suggested by the reviewer. The local validation with FLUXNET has, as recognized also by reviewer #2 been done in Smith et al. [2020] and we moreover highlight the detailed comparison of SiB4 photosynthesis at site-level in Kooijmans et al. [2021], while the integrated validation with atmospheric data is presented in the manuscript. So we focus first on the suggested approach (3), before expanding the local analysis with EC-data that we started in Smith et al. [2020], Kooijmans et al. [2021].

(3) The reviewer suggests to compare to FLUXCOM and Zeng et al. [2020] at regional level. We therefore analysed the PFT-aggregated mean of the net ecosystem productivity (NEP) from both FLUXCOM, Zeng et al. [2020] and our CTE-HR product for the growing season (MJJA) of 2017. Note that the reviewer suggested to assess midday fluxes. However, as both the standard FLUXCOM product, and the product by Zeng et al. [2020] are not available as hourly products, we opted for an analysis over the growing season. We find that both FLUXCOM and the product by Zeng et al. [2020] have a higher NEP than CTE-HR but that the latter might better agree with regional integrals. Table 1 shows the differences, where the high NEP corresponds to earlier reports [Jung et al., 2020, Zeng et al., 2020] of large NEP (globally integrated near 10 PgC/yr sink in FLUXCOM). It also agrees with a tendency for EC-based analyses to represent high uptake locations rather than lower or average locations, leading to potential overestimates of the machine-learning derived fluxes [Jung et al., 2020, Zeng et al., 2020]. In contrast, fluxes optimized from atmospheric $CO_2$ data from the European ICOS network in CT-Europe Friedlingstein et al. [2022] suggest a lower NEP, with the integral matched by CTE-HR more closely than the other products for most PFTs.

In GPP, which arguably is the more reliable machine-learning product (for FLUXCOM at least), our CTE-HR fluxes however also demonstrate a low tendency, specifically for GPP at needleleaf ecosystems found in Scandinavia and for C3 crops. The latter is a generic PFT in SiB4, and mostly used as placeholder for specific crop species that are part of SiBCrop [Lokupitiya et al., 2009], which CTE-HR does not fully utilize yet as it requires detailed local

|         | ENF   | DNF   | EBF   | DBF    | SHRUB | C3GRASS | C3CROPS | C4CROPS |
|---------|-------|-------|-------|--------|-------|---------|---------|---------|
| *GPP*   |       |       |       |        |       |         |         |         |
| CTE-HR  | 6.40  | 19.98 | 1.24  | 26.44  | 6.57  | 22.28   | 39.52   | 0.47    |
| FLUXCOM | 9.60  | 20.40 | 2.06  | 23.43  | 7.17  | 21.23   | 42.99   | 0.92    |
| Zeng    | 11.36 | 24.92 | 2.68  | 31.58  | 9.64  | 25.62   | 52.49   | 1.21    |
| *NEP*   |       |       |       |        |       |         |         |         |
| CTE-HR  | -1.52 | -4.30 | -0.21 | -5.37  | -1.34 | -4.77   | -10.55  | -0.09   |
| FLUXCOM | -3.04 | -6.19 | -0.58 | -8.02  | -2.13 | -6.87   | -14.28  | -0.19   |
| Zeng    | -3.23 | -7.56 | -0.73 | -10.45 | -2.95 | -7.42   | -15.60  | -0.29   |
| CTE     | -2.48 | -4.09 | -0.76 | -6.32  | -1.94 | -5.44   | -14.12  | -0.29   |

**Table 1:** Gross primary production (GPP) and Net Ecosystem Productivity (NEP), integrated over the growing season (MJJA) of 2017 in TgC/month for different land-use types (ENF: evergreen needleleaf forests; DNF: deciduous needleleaf forests; EBF: evergreen broadleaf forests; DBF: deciduous broadleaf forests; SHRUB: Shrublands; C3GRASS: C3 grasslands; C3Crops: C3 croplands; C4CROPS: C4 croplands. CTE refers to NEP optimized using atmospheric data, and published in Friedlingstein et al. [2022]

data on local sowing, irrigation, harvesting, and crop selection. The agreement with FLUXCOM is generally better than with Zeng et al. [2020], and CTE-HR typically is slightly low on GPP. The differences for both NEP and GPP are generally within the "local" error based on EC-data that we will present below.

The results of this analysis are now discussed in our paper (L.496) for further context.

(1) We compared the CTE-HR biosphere flux to the observed NEP at 17 flux towers, in a 0.5° by 0.5° area around the tower where the land-use class is similar to the plant functional type that we use for the downscaling. Following the reviewer's suggestion, we analyse the daytime mean fluxes (between 11h and 16h local time). We assess the Mean Error (ME, the bias), the root mean square error (RMSE) and the correlation coefficient (R). As these metrics vary over the year, we denote them per season. The 17 used towers are: 'SE-Htm', 'BE-Bra', 'FI-Hyy', 'DK-Vng', 'DE-RuS', 'SE-Svb', 'FR-Bil', 'DE-Tha', 'BE-Vie', 'FR-Fon', 'SE-Nor', 'CH-Dav', 'DE-Geb', 'DE-HoH', 'BE-Lon', 'FR-Lam', 'IT-SR2' [Arriga et al., 2022, Brut et al., 2022, Heinesch et al., 2022, Rebmann et al., 2022, Bruemmer et al., 2022, Buchmann et al., 2022, Mölder et al., 2022, Dufrêne et al., 2022, Vincke et al., 2022, Bernhofer et al., 2022, Loustau et al., 2022, Peichl et al., 2022, Schmidt et al., 2022, Friborg et al., 2022, Mammarella et al., 2022, Janssens et al., 2022, Heliasz et al., 2022]

Generally, we underestimate the magnitude of the locally observed biosphere flux with our high-resolution flux product. This is shown in Table 2. This low bias also is reflected in relatively large RMSE values. From the table, we also see that CTE-HR NEP does not capture the day-to-day variability during daytime conditions well during winter in deciduous broadleaf forests and in autumn in croplands, with low correlation coefficients. Note however, that we compare the local eddy-covariance tower signals to a simulated area of about 15x15km here, and we also do not use tower-based driver data for our flux model, which has been shown to lead to large improvements in such evaluations. However, for an error estimate on the 15x15km gridscale we feel that the current evaluation of the *gridded* CTE-HR fluxes is more suited, despite its lower agreement. We have added this table to the Appendix of the revised manuscript, and point to it in the discussion, in L. 510.

For a validation of the downscaling of the SiB4 product, we point the reviewer to our answer to reviewer #2, point 2.

| Land-use type | Winter (DJF) | | | Spring (MAM) | | | Summer (JJA) | | | Autumn (SON) | | |
|---|---|---|---|---|---|---|---|---|---|---|---|---|
| | ME | RMSE | R | ME | RMSE | R | ME | RMSE | R | ME | RMSE | R |
| Evergreen Needleleaf Forests (N=10) | -1.33 | 2.49 | 0.54 | -5.24 | 6.20 | 0.70 | -4.49 | 5.63 | 0.42 | -3.81 | 4.87 | 0.74 |
| Croplands (N=5) | -1.70 | 2.58 | 0.33 | -7.75 | 15.10 | 0.48 | -7.10 | 15.06 | 0.08 | -4.02 | 5.36 | 0.19 |
| Deciduous Broadleaf Forests (N=2) | -0.07 | 1.48 | -0.07 | -3.15 | 6.76 | 0.81 | -12.28 | 13.10 | 0.50 | -4.97 | 7.26 | 0.79 |

**Table 2:** Mean error (ME, $\mu mol/m^2/s$), root mean square error (RMSE, $\mu mol/m^2/s$) and Pearson's correlation coefficient (R) of daytime means (11-16h) of the CTE-HR biosphere fluxes, compared to flux towers (N=17) at similar land-use types, for different seasons. The number provided is the median of the N sites per PFT (in parentheses)

**1.3 Technical corrections.**

- *L1 'We present the CarbonTracker Europe High-Resolution system'. General reader familiar with CTE would suspect the CTE-HR is an inverse modelling system like CTE, but here the name CTE-HR is given to a set of [prior] fluxes, so it is better to explain the difference in the abstract, eg write something like: 'We present the CarbonTracker Europe High-Resolution system fluxes', and note that these fluxes are unoptimized (either in the abstract or in the text, eg lines 115-118)*

We agree with the reviewer, and thank them for noting this. We changed the text to read: "The CTE-HR framework uses the CarbonTracker Data Assimilation Shell (CTDAS) framework to provide near real time flux estimates. These are not optimized with atmospheric data and can be used stand-alone, or they can later be optimized using CTE or other similar data assimilation system." in line 119.

- *L49 For Paris example (Breon et al 2015), one can cite a recent paper by Nalini et al 2022*

We thank the reviewer for this suggestion and changed the reference.

- *L54 N2 -¿ N2O. In case of Swiss methane emissions, Henne et al 2016 is more widely cited.*

We have added this reference.

- *L86-89 As for high resolution, operational NRT biospheric flux products one can mention SMAP L4C (Jones et al 2017)*

We thank the reviewer for this addition and have added the reference.

- *L96 CAMS is associated with a wide variety of products related to the topic of this paper, better give more specific name like CAMS-REG. Also, the paper by Kuenen et al 2022 appears to document CAMS-REG-AP (air pollutant) inventory, while GHG portion called CAMS-REG-GHG was set aside.*

We agree with the addition of a specific name, and have added it (CAMS-REG-GHG). Kuenen et al. [2022] indeed mainly documents the air pollutants, but the DOI cited in the paper also points to the greenhouse gas data. Nevertheless, we have also added the specific dataset DOI (doi:10.24380/0vzb-a387)

- *L168-184 It is not clear, what data is used for spatial emission patterns.*

We agree with the reviewer. For each sector, we added a sentence on the spatial distribution.

- *L215 Final revised paper for GFAS is Di Guiseppe et al. 2018 (https://doi.org/10.5194/acp-18-5359-2018)*

We thank the reviewer for their suggestion, and change the reference.

- *L231 Can cite here the (Chevallier et al 2019) method as 'poor man's inversion' as done by Chevallier et al. (2010, 2019)*

We understand the comment by the reviewer. Currently, we cite Chevallier et al. [2009] as the 'poor persons inversion'. We chose this gender neutral terminology purposefully, as we think it is more inclusive.

- *L249 "Emissions by" can be omitted.*

We omitted 'Emissions by', and thank the reviewer for their suggestion.

- *L255 'exact monthly growth rate' may draw doubts, as the rate is not that exact, can write 'exactly follow monthly growth rate' instead.*

We agree with the reviewer that this formulation was unclear. We thank the reviewer for their suggestion, and changed the line.

- *L280 Need to add spatial/vertical resolution at which IFS winds are used, and STILT domain geographical boundaries.*

We agree with the reviewer. We added the requested information in the text.

- *L471 Suggest clarifying the text: "Currently, CTE-HR only provides CO2 fluxes". Better be more specific: eg biogenic/biospheric/net . . . .*

We agree with the reviewer, and adjusted the text to include the flux components we provide.

- *L524 Here, fluxes 'will be made available ', while on L518 same fluxes 'are available'*

We agree with the reviewer, and have adjusted the data availability statement (L524).

**2 Reviewer 2 (anonymous)**

*The study by van der Woude et al. (2022) described the CarbonTraker Europe High-Resolution (CT-HR) system that estimates carbon dioxide exchange over Europe at high-resolution (0.1 x0.2) in near real-time (about 2-months latency). The system includes a dynamic fossil fuel emission model, hourly net biosphere exchange (NEE), and an ocean flux. All these flux components are hourly. The high spatiotemporal resolution is realized by incorporating easily available statistics on economic activity, energy use, weather, and land cover map. The study evaluated the CT-HR in several aspects, including: 1) comparing the simulated CO2 to atmosphere CO2 observations at ICOS sites; 2) analyzing the 2018 Europe drought signal; 3) analyzing fossil fuel emission anomalies due to the COVID lockdown; and 4) comparing the hourly fluxes against observations from a flux tower at a city center. Overall, the paper is well written. The low-latency high-resolution data will facilitate the high-resolution atmospheric flux inversions, and the near-real time aspect can provide rapid information on carbon cycle changes over Europe. My main concern is the uncertainty quantification and the evaluation of the high spatiotemporal resolution aspect of the product. Please see my detailed comments below.*

We thank the anonymous reviewer for their feedback and constructive comments, which help to improve the paper. Below, we address the comments one by one, with the reviewer comments in italic.

**2.1 Detailed comments:**

- *For each of the flux components, it would be critical to have uncertainty estimation in order to use them in the atmospheric flux inversions and provide useful information on carbon cycle changes, which are the two applications envisioned by the authors. To some degree, the uncertainty is as important as the absolute flux estimation itself for atmospheric flux inversion application. So I would suggest adding uncertainty estimation in the product that propagate the uncertainty of the input data to the uncertainty of the end products.*

We agree that reliable uncertainty estimates would be a very welcome addition to CTE-HR. In fact, such reliable estimates form somewhat of a holy grail in the field, to help inform inversion and machine-learning methodologies and to exploit the large amount of expected satellite observations in the coming years. However, they are notoriously difficult to provide, as documented in several recent publications (Super et al. [2020a], Jung et al. [2020], Liu et al. [2020], Elguindi et al. [2020] and Solazzo et al. [2021]). In our efforts we ran into similar limitations (e.g. a lack of uncertainty estimates from underlying data providers, missing information to make spatial correlations, temporal downscaling based on proxies with non-linear relations to emissions, etc). Therefore, in the manuscript, we only provided an uncertainty estimate in the discussion (L.430), and we based our values on the work of Super et al. [2020a] and Liu et al. [2020] with which we share our methodology, at least for the anthropogenic emissions. This means we do not provide a gridded uncertainty estimate for each hour and each gridbox that we provide fluxes on, but rather stick to integrated spatiotemporal scales. We believe that a gridded hourly uncertainty that remains meaningful when scaling up to the original input data is beyond our capacity to provide.

To accommodate the reviewers request, we made an effort to expand our aggregated uncertainty analysis of the biosphere fluxes, and we clarified our approach to estimate the anthropogenic flux uncertainty as discussed in the manuscript.

**Biosphere flux uncertainty** For the biosphere flux uncertainty, we compare our SiB4 biosphere fluxes to eddy-covariance (EC) measurements. For details on this analysis, we point the reviewer to our reply to reviewer #1 and below we just repeat the resulting table with the metrics, that we added to the Appendix of the manuscript. Like the EC data itself, this analysis provides a spatially sparse set of error estimates, but a spatial representation can be obtained by applying the mean percentage error (MPE, the ratio of the mean error to the absolute NEP) of each PFT to all gridcells with that PFT. Here, we also include nighttime fluxes as well by analysing the MPE of

the daily mean NEP, as we expect people to also transport nighttime fluxes. The MPE is shown in Figure 1 of this rebuttal.

| Land-use type | Winter (DJF) | | | Spring (MAM) | | | Summer (JJA) | | | Autumn (SON) | | |
|---|---|---|---|---|---|---|---|---|---|---|---|---|
| | ME | RMSE | R | ME | RMSE | R | ME | RMSE | R | ME | RMSE | R |
| Evergreen Needleleaf Forests (N=10) | -1.33 | 2.49 | 0.54 | -5.24 | 6.20 | 0.70 | -4.49 | 5.63 | 0.42 | -3.81 | 4.87 | 0.74 |
| Croplands (N=5) | -1.70 | 2.58 | 0.33 | -7.75 | 15.10 | 0.48 | -7.10 | 15.06 | 0.08 | -4.02 | 5.36 | 0.19 |
| Deciduous Broadleaf Forests (N=2) | -0.07 | 1.48 | -0.07 | -3.15 | 6.76 | 0.81 | -12.28 | 13.10 | 0.50 | -4.97 | 7.26 | 0.79 |

**Table 3:** Mean error (ME, $\mu mol/m^2/s$), root mean square error (RMSE, $\mu mol/m^2/s$) and Pearson's correlation coefficient (R) of daytime means (11-16h) of the CTE-HR biosphere fluxes, compared to flux towers (N=17) at similar land-use types, for different seasons. The number provided is the median of the N sites per PFT (in parentheses)

This figure mostly illustrates that the gridded uncertainty on the biospheric fluxes can become very large, if one assumes there are no cancelling errors (error covariances) in space and time. In reality, NEP errors in one gridbox are likely to be compensated by errors in other fluxes, as small-scale variations in the environmental driver data cancel on larger scales of synoptic systems. For example, the passage of a cold-front can create a large error in space and time if the event occurs one hour too late, and 20km too far east in SiB4, severely hurting a comparison to the EC-data. On the larger scale though an equally large and anti-correlated error would occur close by, allowing the simulated NEP when integrated over the full day and multiple locations to be a robust representation of the cold-front passage event. This underlines the difficulty of providing grid-cell based uncertainties, as well as the derivation of NEE-uncertainties based on EC-data. The latter was discussed extensively also in Chevallier et al. [2006].

**Anthropogenic fluxes** The anthropogenic fluxes we provide are partly rooted in the methodology we designed in Super et al. [2020b]. A similar methodology was subject to an extensive uncertainty analysis in Super et al. [2020a]. This analysis suggested that the scaling of country total emissions (uncertainty of ±2%) down to the gridcell-level (1x1 km) increased the yearly uncertainty to 18% of the flux, assuming a Gaussian error distribution. Similarly, the analysis of Liu et al. [2020] suggested an increase in uncertainty to 7% of the flux when extrapolating and scaling from the annual country total, to the daily total. If we assume the spatial scaling error to our 15x15km grid to also be 18% (a possibly somewhat high estimate), and for this error to be independent of the temporal scaling error of 7%, their addition in quadrature yields a total daily gridcell uncertainty of 19.3% of the calculated flux. This number is provided and explained in the text, and is our best estimate for the daily uncertainty on our gridded fluxes. We extended the discussion to be more explicit (L.466; L.494) on these uncertainties.

- *The evaluation of the high spatiotemporal feature of the product is not sufficient. The paper underwent extensive effort to evaluate the product, but the evaluation is mainly on large spatial and monthly time scale (except the tower comparison and partially the ICOS comparison), which are actually mainly from the input dataset. For example, the 2018 drought was evaluated on an aggregated region, the COVID signal was also evaluated on the continental scale by comparing to the published results. As the high spatiotemporal feature is the main advancement, I would recommend adding more evaluation at finer spatiotemporal scales. For example, the observations from flux towers can be used to evaluate the hourly NEP from SIB4.*

We thank the reviewer for acknowledging the extensive effort we put into the validation of our product, and we agree with the reviewer that specifically the high-resolution validation of CTE-HR can be elaborated. Therefore, we validate two important components: the added value of (a) the *daily* ENTSO-E data on power-usage and b) the *15km spatial downscaled* SiB4 biosphere fluxes.

[Figure]

**Figure 1:** Mean percentage error (MPE) of daily mean NEP between the downscaled SiB4 fluxes and eddy-covariance towers over Europe. The applied percentage is derived from the comparison to daily means of hourly EC-data, following the methodology described in our answer to Reviewer #1, point 1. The figure is based on data from 17 sites, with measurements from 2017 to 2021, and thus represents a temporal aggregation of our flux product. Missing data means that no flux towers are available over that land-cover type. Note that the colourbar in autumn is cut-off for readability, and that the MPE of deciduous broadleaf forests (DBF) in the autumn is -887%, owing to the small NEP measured (-0.1 $\mu mol/m^2/s$).

**Daily ENTSO-E data**   The main added value of ENTSO-E power usage data -relative to monthly data used widely in the community- shows in specific cases where deviations from the mean flux is large. An example of this occurred during Christmas 2017 in Germany when, due to high wind speeds and an abundance of sustainable energy, German electricity prices went negative. This event was widely covered in media (e.g. `https://www.businessinsider.com/renewable-power-germany-negative-electricity-cost-2017-12?international=true&r=US&IR=T`). With the ENTSO-E data, our CTE-HR emissions capture this increase in wind-generated electricity, and the corresponding decrease in energy generation by the combustion of fossil coal and gas. This is shown in the Figure 2, demonstrating the added value over a lower temporal resolution view such as provided by the CAMS emission dataset. We emphasize that the difference between the public power flux according to CAMS-REG-GHG emissions and the ENTSO-E data is about 1/3rd of the biosphere fluxes in Germany during December, and an error due to monthly constant anthropogenic emissions is thus unlikely to be corrected using atmospheric in-situ or space-based data. Instead, the reduced emissions from power generation would be wrongly attributed to the biosphere fluxes, or to other larger emissions sources if no sub-monthly data on power generation was available in the underlying emissions. This we see as a clear added value of higher temporal resolution in our product. We have added this explanation and figure to the updated manuscript (L.253).

**Added value of the spatial downscaling of SiB4**   Our downscaling from 0.5 degrees SIB4 output to ˜15km (0.1x0.2 degrees) resolution in CTE-HR similarly adds useful information to the spatial distribution of the biosphere fluxes. This is shown in Figure 3, where we show the spatial correlation coefficient between GPP fluxes in CTE-HR, and MODIS-derived Near-Infrared Reflection of vegetation (NIRv, see Badgley et al. [2019]), both at 0.05 degrees (CTE-HR regridded using nearest-neighbour resampling). The correlation is calculated over N=100 high-resolution pixels falling within larger 0.5 by 0.5° boxes for July 2016. A positive correlation coefficient between observed NIRv and simulated GPP suggests that credible sub-0.5 degree gradients were present in the high-resolution CTE-HR fluxes, even though they were originally calculated at a coarser (0.5 degree) resolution in SiB4. We find that for 1795 (56%) of the larger gridboxes, the spatial downscaling indeed represents the observed gradient in a statistically significant correlation. For 507 larger gridboxes (16%), the observed spatial gradient is misrepresented, and for 894 boxes (28%), no conclusions can be drawn due to lack of observed variability or lack of a significant correlation within the 0.5 by 0.5° gridbox. This result was found similar for other months in the growing season (when GPP is higher), and demonstrates that the upscaled biosphere fluxes are in the majority of boxes better than (54%), or at least as good as (28%), the SIB4 fluxes without downscaling. We have also added this figure to the text of the updated manuscript (L.240).

- *It is not clear how the diurnal cycle of SIB4 was calculated, and what is the temporal frequency of SIB4 output. Is the original SIB4 output monthly or daily?*

The original SiB4 output is hourly. We stated this more clearly in the updated manuscript (L. 204).

- *Figure 6 shows that the HR almost has the same mean RMSE as the PRI, but the text claims that the CT-HR performs better. It would be informative to include the exact RMSE from PRI and CT-HR when comparing to ICOS network in the text.*

We agree with the reviewer, and have put the median RMSE, which confirms our original statement, in the text near the figure.

- *Figure 7 shows that all three fluxes give very similar results during well-mixed hours, but it is hard to tell which one has better performance at the bottom two panels since the scale is so large. I would suggest adding panels to show the difference between the simulated run and the measurements. I would also suggest discussing RMSE of CO2 forced by these three fluxes during well-mixed hours and during night and early morning.*

[Figure]

**Figure 2:** Estimated $CO_2$ emission from the energy sector, according to the ENTSO-E reported data, recalculated following the method described in the manuscript (blue) and the CAMS-REG-GHG dataset (orange) over Germany for December 2017. The grey shading indicates the period in which the negative energy prices occurred is indicated

[Figure]

**Figure 3:** Correlation coefficient between MODIS NIRv and the downscaled SiB4 product from CTE-HR (both at 0.05° by 0.05°) within 0.5x0.5 degree gridcells (i.e., N=100 per box shown). Non-significant spatial correlation coefficients are marked with a cross. The inset in the upper right shows the amount of gridcells where the spatial gradient was well-represented or misrepresented, or where no significant spatial correlation between observations and CTE-HR model was found.

Following the suggestion by the reviewer, we added two subplots, however, we found that these made the figure more complicated, whilst adding limited information. We therefore opt to keep the figure as is, and would like to restate that the differences are simply small, and with the current transport model (STILT, driven by IFS), the added value of the high-resolution fluxes is hard to show.

We agree with the reviewer that adding a discussion on the RMSE during different stability regimes is interesting, and we have added such a discussion to the text (L.402).

- *It is strange that the "Flat" fluxes seem to have the same performance as the "full res" in December, which seem to indicate potential large errors in boundary layer height. I would suggest evaluating boundary layer height information used in STILT and isolate the impact of transport and fluxes on the flux errors shown in Figure 7.*

We apologize for possibly not understanding fully the point of the reviewer here. The 'Flat' and 'full res' fluxes

have indeed about the same performance in December, when the biosphere is dormant. Therefore, there is a lack of a diurnal cycle from the biosphere fluxes in December, which is reflected in the similar performance for 'Flat' and 'full res'. However, we do understand the concerns of the reviewer regarding the boundary layer height, despite its simulations being a known issue for Lagrangian transport models, such as the used STILT model [Maier et al., 2022].

Although our paper does not rely heavily on the use of Lagrangian modeling except for the Figure remarked on, we attempt to address a BLH-concern for our study. To quantify a potential error in the boundary layer height, we compared the mixed layer height, as used by STILT (from IFS) with the boundary layer height taken from ERA5 reanalysis data (which is also known to have uncertainties, e.g. Sinclair et al. [2022]). This analysis showed that the mixed layer height, used in STILT, is structurally 200m lower than the boundary layer height as simulated by ERA5. Assuming that all emissions are well-mixed through the boundary layer in STILT, we can correct for differences in the boundary layer height, by multiplying the enhancement due to $CO_2$ fluxes by the ratio between the boundary layer heights. This linear correction results in simulated mole fractions of about 2.5ppm lower when using the ERA5 boundary layer height, instead of the mixed layer height from IFS. Although such an offset is highly relevant for modeling absolute levels of $CO_2$, a 200m *systematic* error would not translate to different temporal gradients, or differences between the flux scenarios in Figure 7. For that, day-to-day variations in BLH errors are more relevant, for which we presented an extensive analysis in Pino et al. [2012]. We furthermore note that we do not have mixing layer height measurements for the observation station of Lutjewad, and the impact of the transport on observed atmospheric mole fractions cannot be isolated from the impact of the fluxes. Although we cannot isolate these impacts, we did point to this issue in the discussion (L.528), and thank the reviewer for pointing out this issue.

**References**

N. Arriga, I. Goded, G. Manca, and ICOS Ecosystem Thematic Centre. Warm winter 2020 ecosystem eddy covariance flux product from San Rossore 2, 2022. URL https://meta.icos-cp.eu/objects/bqA7N1QkO90Oz0QqzsUWsKnK.

G. Badgley, L. D. Anderegg, J. A. Berry, and C. B. Field. Terrestrial gross primary production: Using NIRV to scale from site to globe. *Global Change Biology*, 25(11):3731–3740, 2019. ISSN 13652486. doi: 10.1111/gcb.14729. URL https://eartharxiv.org/s6t3z/.

C. Bernhofer, J. T. Gruenwald, and ICOS Ecosystem Thematic Centre. Warm winter 2020 ecosystem eddy covariance flux product from Tharandt, 2022. URL https://meta.icos-cp.eu/objects/-kQhLRmom_xBsI_1nZISHXvN.

C. Bruemmer, F. Schrader, J.-P. Delorme, A. Lucas-Moffat, and ICOS Ecosystem Thematic Centre. Warm winter 2020 ecosystem eddy covariance flux product from Gebesee, 2022. URL https://meta.icos-cp.eu/objects/_9OUPlOC-LChCxEpZMmdg-00.

A. Brut, T. Tallec, E. Ceschia, and ICOS Ecosystem Thematic Centre. Warm winter 2020 ecosystem eddy covariance flux product from Lamasquere, 2022. URL https://meta.icos-cp.eu/objects/-4TMRFcN7N-iM4cR3MDSbeq0.

N. Buchmann, L. Hörtnagl, L. Merbold, M. Gharun, and ICOS Ecosystem Thematic Centre. Warm winter 2020 ecosystem eddy covariance flux product from Davos, 2022. URL https://meta.icos-cp.eu/objects/sFaiiz5zR6xQptdXMmyKmuok.

F. Chevallier, N. Viovy, M. Reichstein, and P. Ciais. On the assignment of prior errors in Bayesian inversions of CO ¡sub¿2¡/sub¿ surface fluxes. *Geophysical Research Letters*, 33(13):L13802, 7 2006. ISSN 0094-8276. doi: 10.1029/2006GL026496. URL http://doi.wiley.com/10.1029/2006GL026496.

F. Chevallier, R. J. Engelen, C. Carouge, T. J. Conway, P. Peylin, C. Pickett-Heaps, M. Ramonet, P. J. Rayner, and I. Xueref-Remy. AIRS-based versus flask-based estimation of carbon surface fluxes. *Journal of Geophysical Research*, 114(D20), 10 2009. ISSN 0148-0227. doi: 10.1029/2009jd012311.

E. Dufrêne, D. Berveiller, N. Delpierre, and ICOS Ecosystem Thematic Centre. Warm winter 2020 ecosystem eddy covariance flux product from Fontainebleau-Barbeau, 2022. URL https://meta.icos-cp.eu/objects/wEkQAxJYVZmr-NQHAVbUWGmM.

N. Elguindi, C. Granier, T. Stavrakou, S. Darras, M. Bauwens, H. Cao, C. Chen, H. A. Denier van der Gon, O. Dubovik, T. M. Fu, D. K. Henze, Z. Jiang, S. Keita, J. J. Kuenen, J. Kurokawa, C. Liousse, K. Miyazaki, J. F. Müller, Z. Qu, F. Solmon, and B. Zheng. Intercomparison of Magnitudes and Trends in Anthropogenic Surface Emissions From Bottom-Up Inventories, Top-Down Estimates, and Emission Scenarios. *Earth's Future*, 8(8), 8 2020. ISSN 23284277. doi: 10.1029/2020EF001520.

T. Friborg, R. Jensen, R. Jensen, and L. Rasmussen. ETC L2 Fluxnet (half-hourly), Voulundgaard, 2019-12-31–2021-12-31, 2022. URL https://hdl.handle.net/11676/lE2EnZqqarSmqny87R8dSNzh.

P. Friedlingstein, M. O'Sullivan, M. W. Jones, R. M. Andrew, L. Gregor, J. Hauck, C. Le Quéré, I. T. Luijkx, A. Olsen, G. P. Peters, W. Peters, J. Pongratz, C. Schwingshackl, S. Sitch, J. G. Canadell, P. Ciais, R. B. Jackson, S. R. Alin, R. Alkama, A. Arneth, V. K. Arora, N. R. Bates, M. Becker, N. Bellouin, H. C. Bittig, L. Bopp, F. Chevallier, L. P. Chini, M. Cronin, W. Evans, S. Falk, R. A. Feely, T. Gasser, M. Gehlen, T. Gkritzalis,

L. Gloege, G. Grassi, N. Gruber, O. Gurses, I. Harris, M. Hefner, R. A. Houghton, G. C. Hurtt, Y. Iida, T. Ilyina, A. K. Jain, A. Jersild, K. Kadono, E. Kato, D. Kennedy, K. Klein Goldewijk, J. Knauer, J. I. Korsbakken, P. Landschützer, N. Lefèvre, K. Lindsay, J. Liu, Z. Liu, G. Marland, N. Mayot, M. J. McGrath, N. Metzl, N. M. Monacci, D. R. Munro, S.-I. Nakaoka, Y. Niwa, K. O'Brien, T. Ono, P. I. Palmer, N. Pan, D. Pierrot, K. Pocock, B. Poulter, L. Resplandy, E. Robertson, C. Rödenbeck, C. Rodriguez, T. M. Rosan, J. Schwinger, R. Séférian, J. D. Shutler, I. Skjelvan, T. Steinhoff, Q. Sun, A. J. Sutton, C. Sweeney, S. Takao, T. Tanhua, P. P. Tans, X. Tian, H. Tian, B. Tilbrook, H. Tsujino, F. Tubiello, G. R. van der Werf, A. P. Walker, R. Wanninkhof, C. Whitehead, A. Willstrand Wranne, R. Wright, W. Yuan, C. Yue, X. Yue, S. Zaehle, J. Zeng, and B. Zheng. Global Carbon Budget 2022. *Earth System Science Data*, 14(11):4811–4900, 11 2022. doi: 10.5194/essd-14-4811-2022.

B. Heinesch, A. De Ligne, T. Manise, B. Longdoz, and ICOS Ecosystem Thematic Centre. Warm winter 2020 ecosystem eddy covariance flux product from Lonzee, 2022. URL `https://meta.icos-cp.eu/objects/URIAsjYz5-1CoewJ3zzNF59K`.

M. Heliasz, J. Holst, and ICOS Ecosystem Thematic Centre. Warm winter 2020 ecosystem eddy covariance flux product from Hyltemossa, 2022. URL `https://meta.icos-cp.eu/objects/YJkrKWJx317TTXKmMk4XZRif`.

I. Janssens, T. De Meulder, M. Roland, J. Segers, and ICOS Ecosystem Thematic Centre. Warm winter 2020 ecosystem eddy covariance flux product from Brasschaat, 2022. URL `https://meta.icos-cp.eu/objects/5BCT4nKCoGaYQh77DW6OW7gs`.

M. Jung, C. Schwalm, M. Migliavacca, S. Walther, G. Camps-Valls, S. Koirala, P. Anthoni, S. Besnard, P. Bodesheim, N. Carvalhais, F. Chevallier, F. Gans, D. S Goll, V. Haverd, P. Köhler, K. Ichii, A. K Jain, J. Liu, D. Lombardozzi, J E M S Nabel, J A Nelson, M. O'Sullivan, M. Pallandt, D. Papale, W. Peters, J. Pongratz, C. Rödenbeck, S. Sitch, G. Tramontana, A. Walker, U. Weber, and M. Reichstein. Scaling carbon fluxes from eddy covariance sites to globe: Synthesis and evaluation of the FLUXCOM approach. *Biogeosciences*, 17 (5):1343–1365, 3 2020. ISSN 17264189. doi: 10.5194/bg-17-1343-2020.

L. M. Kooijmans, A. Cho, J. Ma, A. Kaushik, K. D. Haynes, I. Baker, I. T. Luijkx, M. Groenink, W. Peters, J. B. Miller, J. A. Berry, J. Ogée, L. K. Meredith, W. Sun, K. M. Kohonen, T. Vesala, I. Mammarella, H. Chen, F. M. Spielmann, G. Wohlfahrt, M. Berkelhammer, M. E. Whelan, K. Maseyk, U. Seibt, R. Commane, R. Wehr, and M. Krol. Evaluation of carbonyl sulfide biosphere exchange in the Simple Biosphere Model (SiB4). *Biogeosciences*, 18(24):6547–6565, 12 2021. ISSN 17264189. doi: 10.5194/bg-18-6547-2021.

J. Kuenen, S. Dellaert, A. Visschedijk, J.-P. Jalkanen, I. Super, and H. Denier van der Gon. CAMS-REG-v4: a state-of-the-art high-resolution European emission inventory for air quality modelling. *Earth System Science Data*, 14(2):491–515, 2 2022. ISSN 1866-3516. doi: 10.5194/essd-14-491-2022. URL `https://essd.copernicus.org/articles/14/491/2022/`.

Z. Liu, P. Ciais, Z. Deng, R. Lei, S. J. Davis, S. Feng, B. Zheng, D. Cui, X. Dou, B. Zhu, R. Guo, P. Ke, T. Sun, C. Lu, P. He, Y. Wang, X. Yue, Y. Wang, Y. Lei, H. Zhou, Z. Cai, Y. Wu, R. Guo, T. Han, J. Xue, O. Boucher, E. Boucher, F. Chevallier, K. Tanaka, Y. Wei, H. Zhong, C. Kang, N. Zhang, B. Chen, F. Xi, M. Liu, F. M. Bréon, Y. Lu, Q. Zhang, D. Guan, P. Gong, D. M. Kammen, K. He, and H. J. Schellnhuber. Near-real-time monitoring of global CO2 emissions reveals the effects of the COVID-19 pandemic. *Nature Communications*, 11 (1), 12 2020. ISSN 20411723. doi: 10.1038/s41467-020-18922-7.

E. Lokupitiya, S. Denning, K. Paustian, I. Baker, K. Schaefer, S. Verma, T. Meyers, C. J. Bernacchi, A. Suyker, and M. Fischer. Incorporation of crop phenology in Simple Biosphere Model (SiBcrop) to improve land-atmosphere carbon exchanges from croplands. *Biogeosciences*, 6:969–986, 2009. URL `www.biogeosciences.net/6/969/2009/`.

D. Loustau, C. Chipeaux, and ICOS Ecosystem Thematic Centre. Warm winter 2020 ecosystem eddy covariance flux product from Bilos, 2022. URL `https://meta.icos-cp.eu/objects/9TbbJgk29oNc2pyEZtvgAF5a`.

F. Maier, C. Gerbig, I. Levin, I. Super, J. Marshall, and S. Hammer. Effects of point source emission heights in WRF-STILT: a step towards exploiting nocturnal observations in models. *Geoscientific Model Development*, 15(13):5391–5406, 2022. ISSN 19919603. doi: 10.5194/gmd-15-5391-2022. URL `https://doi.org/10.5194/gmd-2021-386`.

I. Mammarella, T. Vesala, P. Kolari, and ICOS Ecosystem Thematic Centre. Warm winter 2020 ecosystem eddy covariance flux product from Hyytiälä, 2022. URL `https://meta.icos-cp.eu/objects/yT6Yil9juot9Gffo111sBous`.

M. Mölder, H. Lankreijer, F. Lagergren, J. Holst, and ICOS Ecosystem Thematic Centre. Warm winter 2020 ecosystem eddy covariance flux product from Norunda, 2022. URL `https://meta.icos-cp.eu/objects/f1tLJAI708YA5w37gX6UNQbY`.

M. Peichl, M. Ottosson Lofvenius, M. B. Nilsson, P. Marklund, and ICOS Ecosystem Thematic Centre. Warm winter 2020 ecosystem eddy covariance flux product from Svartberget, 2022. URL `https://meta.icos-cp.eu/objects/ujM7vxk12vKCoquLSToLG18z`.

D. Pino, J. Vilà-Guerau De Arellano, W. Peters, J. Schröter, C. C. Van Heerwaarden, and M. C. Krol. A conceptual framework to quantify the influence of convective boundary layer development on carbon dioxide mixing ratios. *Atmospheric Chemistry and Physics*, 12(6):2969–2985, 2012. ISSN 16807316. doi: 10.5194/acp-12-2969-2012. URL `www.atmos-chem-phys.net/12/2969/2012/`.

C. Rebmann, L. Dienstbach, S. Gimper, and ICOS Ecosystem Thematic Centre. Warm winter 2020 ecosystem eddy covariance flux product from Hohes Holz, 2022. URL `https://meta.icos-cp.eu/objects/BlmyC43RrLhzqezyaEPJcJDs`.

M. Schmidt, A. Graf, D. Dolfus, and ICOS Ecosystem Thematic Centre. Warm winter 2020 ecosystem eddy covariance flux product from Selhausen Juelich, 2022. URL `https://meta.icos-cp.eu/objects/lttDaMaEVOvI2OOC5QEReMic`.

V. A. Sinclair, J. Ritvanen, G. Urbancic, I. Statnaia, Y. Batrak, D. Moisseev, and M. Kurppa. Boundary-layer height and surface stability at Hyytiala, Finland, in ERA5 and observations. *Atmospheric Measurement Techniques*, 15(10):3075–3103, 5 2022. ISSN 18678548. doi: 10.5194/amt-15-3075-2022.

N. E. Smith, L. M. Kooijmans, G. Koren, E. Van Schaik, A. M. Van Der Woude, N. Wanders, M. Ramonet, I. Xueref-Remy, L. Siebicke, G. Manca, C. Brümmer, I. T. Baker, K. D. Haynes, I. T. Luijkx, and W. Peters. Spring enhancement and summer reduction in carbon uptake during the 2018 drought in northwestern Europe: Carbon uptake during 2018 Eur. drought. *Philosophical Transactions of the Royal Society B: Biological Sciences*, 375(1810), 10 2020. ISSN 14712970. doi: 10.1098/rstb.2019.0509. URL `https://royalsocietypublishing.org/doi/abs/10.1098/rstb.2019.0509`.

E. Solazzo, M. Crippa, D. Guizzardi, M. Muntean, M. Choulga, and G. Janssens-Maenhout. Uncertainties in the Emissions Database for Global Atmospheric Research (EDGAR) emission inventory of greenhouse gases. *Atmospheric Chemistry and Physics*, 21(7):5655–5683, 4 2021. ISSN 16807324. doi: 10.5194/acp-21-5655-2021.

I. Super, S. N. Dellaert, A. J. Visschedijk, and H. D. van der Gon. Uncertainty analysis of a European high-resolution emission inventory of CO2 and CO to support inverse modelling and network design. *Atmospheric*

*Chemistry and Physics*, 20(3):1795–1816, 2020a. ISSN 16807324. doi: 10.5194/acp-20-1795-2020. URL `https://doi.org/10.5194/acp-20-1795-2020`.

I. Super, H. A. Denier Van Der Gon, M. K. Van Der Molen, S. N. Dellaert, and W. Peters. Optimizing a dynamic fossil fuel CO2 emission model with CTDAS (CarbonTracker Data Assimilation Shell, v1.0) for an urban area using atmospheric observations of CO2, CO, NOx, and SO2. *Geoscientific Model Development*, 13(6):2695–2721, 6 2020b. ISSN 19919603. doi: 10.5194/gmd-13-2695-2020. URL `https://research.wur.nl/en/publications/optimizing-a-dynamic-fossil-fuel-co2-emission-model-with-ctdas-ca`.

C. Vincke, B. Heinesch, B. Longdoz, and ICOS Ecosystem Thematic Centre. Warm winter 2020 ecosystem eddy covariance flux product from Vielsalm, 2022. URL `https://meta.icos-cp.eu/objects/roUnkgssFKhxi3o9zcIP00Cw`.

J. Zeng, T. Matsunaga, Z. H. Tan, N. Saigusa, T. Shirai, Y. Tang, S. Peng, and Y. Fukuda. Global terrestrial carbon fluxes of 1999–2019 estimated by upscaling eddy covariance data with a random forest. *Scientific Data*, 7(1), 2020. ISSN 20524463. doi: 10.1038/s41597-020-00653-5. URL `www.nature.com/scientificdata`.

---

## Author Response (AR2)

**1   General comments**

- The revised manuscript takes into full consideration the suggestions made by reviewers. The authors did a lot of work and new analysis to implement the recommendations. The manuscript can be accepted after minor revision.

We thank the reviewer for acknowledging our extensive efforts.

**2   Detailed comments**

- L266 Here authors introduce a term 'poor-person's inversion' as a version of 'poor man's inversion' by Chevallier et al (2009). The motivation to rebrand the term 'poor man's something' appears shaky, given the fact the saying itself is deep-rooted in history, and became a part of English language. For example, a newspaper named 'Poor Man's Guardian' was published in 1831-1835 (there is a story in Wikipedia). A suggested replacement has not been as widely accepted, a Google search on it returns little of significance. The recommendation is to stick to the original term as coined by Chevallier et al (2009).

We have added original term to the reference Chevallier et al. [2009]

**References**

F. Chevallier, R. J. Engelen, C. Carouge, T. J. Conway, P. Peylin, C. Pickett-Heaps, M. Ramonet, P. J. Rayner, and I. Xueref-Remy. AIRS-based versus flask-based estimation of carbon surface fluxes. *Journal of Geophysical Research*, 114(D20), 10 2009. ISSN 0148-0227. doi: 10.1029/2009jd012311.